



# Effects of basal drag on subduction dynamics from 2D numerical models

Lior Suchoy[1], Saskia Goes[1], Benjamin Maunder[1], Fanny Garel[2], Rhodri Davies[3]

[1]Department of Earth Science and Engineering, Imperial College of London, South Kensington Campus, London, UK
[2]Géosciences Montpellier, Université de Montpellier, CNRS, Montpellier, France
[3] Research School of Earth Sciences, Australian National University, Canberra, Australian Capital Territory, Australia

*Correspondence to*: Lior Suchoy (l.suchoy17@imperial.ac.uk)

**Abstract.** Subducting slabs are an important driver of plate motions, yet the force balance governing subduction dynamics remains incompletely understood. Basal drag has been proposed to be a minor contributor to subduction forcing, because of
the lack of correlation between plate size and velocity in observed and reconstructed plate motions. Furthermore, in single subduction system models, low basal drag, associated with a low ratio of asthenospheric to lithospheric viscosity, leads to subduction behaviour most consistent with the observation that trench migration velocities are generally low compared to convergence velocities. By contrast, analytical calculations and global mantle flow models indicate basal drag can be substantial. In this study, we revisit this problem by examining the drag at the base of the lithosphere, for a single subduction
system, in 2D models with a free trench and composite non-linear rheology. We compare the behaviour of short and long plates for a range of asthenospheric and lithospheric rheologies. We reproduce results from previous modelling studies, including low ratios of trench over plate motions. However, we also find that any combination of asthenosphere and lithosphere viscosity that produces Earth-like subduction behaviour leads to a correlation of velocities with plate size, due to the role of basal drag. By examining Cenozoic plate motion reconstructions, we find that slab age and plate size are
positively correlated: higher slab pull for older plates tends to be offset by higher basal drag below these larger plates. This, in part, explains the lack of plate velocity-size correlation in observations, despite the important role of basal drag in the subduction force-balance.

## 1 Introduction

It is widely agreed that the negative buoyancy of subducting slabs provides the main driving force for subduction, as well as
for plate motions in general (e.g. Forsyth and Uyeda, 1975; Becker and O'Connell, 2001; Conrad and Lithgow-Bertelloni, 2002; Coltice et al., 2019). This insight has led to first-order models where slabs are treated as Stokes sinkers through a viscous mantle, which successfully produce the main density structure of the mantle and the main patterns of plate motions, assuming that surface plate motions are driven by the resulting mantle flow (e.g. Lithgow-Bertelloni and Richards, 1998; Becker and O'Connell, 2001; Conrad and Lithgow-Bertelloni, 2002). Recent global or plate-scale studies show that in some
cases additional forcing by mantle plumes and background mantle flow (leading to active drag) may also play a role (e.g.,



Becker and Faccenna, 2011; Coltice et al., 2019; Stotz et al., 2018). Nonetheless, such large-scale models cannot fully capture the regional force balance governing the dynamics of subduction at plate boundaries (i.e. the interplay between slab pull, plate bending, viscous forces between the mantle and the plates), in governing plate velocities and morphology. As a result, it is important to improve our understanding of the balance of forces in such a regional system to inform how these

forces, or a parameterisation of them, would be best incorporated into larger-scale or global models (e.g. Conrad and Hager, 1999).

The main driving force in a regional subduction system is the pull of the sinking slab. Resistive forces include viscous drag on the slabs, frictional or viscous resistance at the contact between subducting and overriding plates, visco-plastic resistance of the subducting plates to bending into the trench and when they reach upper-lower mantle boundary (ULMB) at 660 km

depth, and viscous drag on the base of the plate (e.g. Forsyth and Uyeda, 1975; Carlson et al., 1983; Conrad and Hager, 1999). For subducting plates, the latter force, basal drag, is assumed to be dominantly resistive, although we note that it can be a driving force (e.g. Becker and Faccenna, 2011; Stotz et al., 2018). A driving basal drag can occur where mantle flow, driven by the global system of sinking slabs and aided by rising plumes, provides additional forcing on a regional plate configuration (Coltice et al., 2019). We will not consider such external forces here. The regional subduction resistive forces

counteract up to 90% of slab pull, with the largest contribution from viscous resistance between the subducting slabs and the surrounding mantle (e.g. Schellart, 2004; Capitanio et al., 2007). It is debated what the relative importance of the resistive forces is, and how their contributions vary between different subduction zones, and how this might contribute to the variability in subduction zone behaviour (e.g. Conrad et al., 2004; Husson et al., 2012).

It has been noted that present-day plates with large subducting slabs attached do not display a correlation between plate size

and plate velocity, as might be expected if basal drag played a significant role in the subduction force balance (e.g. Forsyth and Uyeda, 1975; Conrad and Hager, 1999; Fig. 1). Conrad and Hager, (1999) suggested that high plate viscosity, which would lead to substantial loss of energy in bending of the subducting plate at the trench (i.e. high bending resistance relative to basal drag), might be responsible for this lack of correlation. Their estimate of effective plate viscosity (about two orders of magnitude higher than the asthenosphere) has been confirmed by a range of subsequent studies (e.g. Zhong and Davies,

1999; Buffett and Rowley, 2006). However, dynamic subduction models with a mobile trench have shown that plates with such effective viscosities tend to adjust their dips and trench velocities to minimise energy lost in bending thus limiting the effect on plate velocities (e.g. Bellahsen et al., 2005; Capitanio et al., 2007; Ribe, 2010).

Other work, using regional dynamic subduction models, has shown that low basal drag resulting from low asthenospheric viscosities may be important for understanding trench motions as observed on Earth (Billen and Arredondo, 2018). Trench

motions tend to be only a small fraction of plate convergence velocity (on average, throughout the Cenozoic, about 10%, Lallemand et al., 2005; Sdrolias and Müller, 2006; Goes et al., 2011). On the other hand, the type of free subduction models used to investigate the effects of plate bending can lead to trench motions that are up to 100% of convergence (Capitanio et al., 2007), although models featuring an overriding plate result in smaller relative trench velocities (Garel et al., 2014). Furthermore, although most trenches retreat, 20-35% of trench segments in the past 100 Myr advanced (Goes et al., 2011;





Williams et al., 2015). Small variations in the balance between the forces affecting the upper and the subducting plates have
been proposed to determine the direction of trench motion (Husson et al., 2012; Capitanio, 2013; Alsaif et al., 2020) and for
such a mechanism to be effective, a low asthenospheric viscosity is likely required. Billen and Arredondo, (2018) showed
that a low viscosity asthenosphere (with non-linear, strain-rate dependent rheology) was important to achieve small trench
motions and periods of both trench advance and retreat.

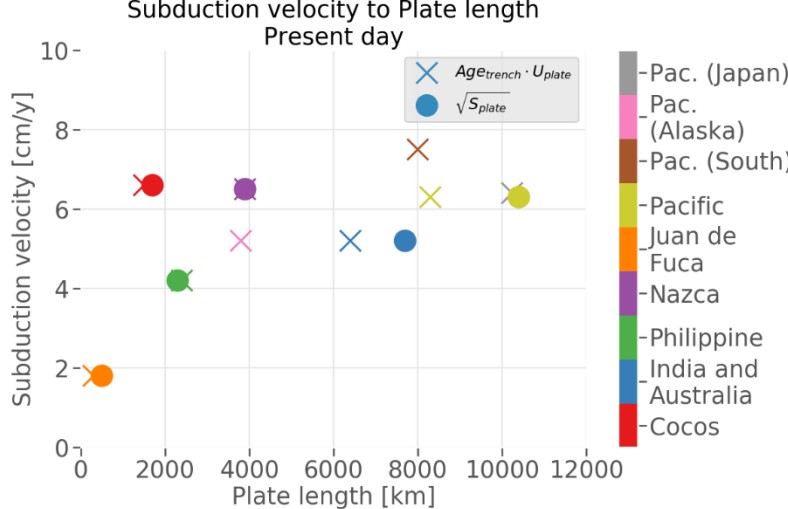


**Figure 1: Velocity of subducting plates at the trench in lower mantle (plume) reference frame (O'Neill et al., 2005), as a function of typical plate length. Plate length calculated as age at the trench ($Age_{trench}$) multiplied by plate velocity ($U_{plate}$) (x markers) or as the square root of the plate surface area ($S_{plate}$, o markers). Data from Conrad and Hager (1999), Sdrolias and Müller (2006) and Schellart et al. (2007).**

Other 2-D models of convection that approximate plate behaviour by a mechanical top boundary condition and strongly
temperature-dependent rheology, albeit with an immobile trench (Höink and Lenardic, 2008, 2010), document how the ratio
of lithosphere to asthenosphere viscosity governs to what extent basal drag is a resistive force. In such models, a low enough
ratio even leads to a driving (Poiseuille) basal drag. From these results, one would expect that if the trench was free to move,
resistive basal drag would lead to trench retreat and driving basal drag would be associated with trench advance, i.e.
viscosity ratio near this transition would lead to small trench motions, as found by Billen and Arredondo (2018). This indeed
appears to be borne out by correlations found by Husson et al. (2012) between modelled mantle flow patterns and trench
motions. Höink and Lenardic (2010) further showed that there is an influence of box size (which simulates plate size in this
work) on the viscosity ratio at which the transition from resisting to driving basal drag occurs. These studies indicate that a
low ratio of average lithosphere to average asthenosphere viscosity, i.e. relatively low-magnitude basal drag, puts the system
close to the transition from retreating to advancing trench motion and hence may be the most appropriate for understanding
natural subduction dynamics.



On the other hand, analytical calculations and global-scale numerical models indicate that basal drag may not be such a negligible force. Basic analytical calculations (Turcotte and Schubert, 2002) yield estimates of basal drag that are on the order of 5-10% of slab pull. Observations of seismic anisotropy (Bokelmann and Silver, 2002) have been used to estimate that drag on the base of thick cratonic lithosphere is of a similar magnitude as ridge push, which is about 10% of slab pull and of a similar magnitude as the net slab pull (i.e. the difference between slab pull and viscous resistance to slab sinking). Several other studies have also shown that basal drag can be high in areas of thick continental roots (e.g. Conrad and Lithgow-Bertelloni, 2006; Coltice et al., 2019) and tractions from global mantle flow models yield high cumulative values of basal drag on large plates like the Pacific (Conrad and Lithgow-Bertelloni, 2006). The magnitude of basal drag by global mantle flow has been shown to be important for explaining plate motions (e.g. Becker and O'Connell, 2001; Conrad and Lithgow-Bertelloni, 2002; Becker and Faccenna, 2011; Coltice et al., 2019) and can only be effective if asthenospheric viscosities are not too low.

To better understand the role that basal drag plays in controlling subduction and trench motions, we revisit these issues here using 2D dynamic subduction models with composite rheology, incorporating several deformation mechanisms with stress-dependent and strongly temperature-dependent rheologies, and mobile trenches. We test the effects of increased lithosphere viscosity, decreased asthenosphere viscosity; hence variable lithospheric to asthenospheric viscosity ratio. Finally, we re-evaluate relation between subducting plate velocity and plate size on Earth, by examining both present-day plate configurations and reconstructions through the Cenozoic Era.

## 2 Model set up

We use a thermo-mechanical model of a single subduction system comprising a subducting and overriding plate in a 2D rectangular box (Fig. 2). The finite-element, control-volume code Fluidity (Davies et al., 2011; Kramer et al., 2012; Garel et al., 2014; Le Voci et al., 2014) is used to solve the mass, momentum and energy conservation equations for Stokes flow under the Boussinesq approximation of incompressibility, for a viscous rheology. An irregular triangular adaptive mesh, with element sizes ranging from 0.4-200 km, yields high resolution in areas of dynamic significance (i.e. in areas of strong curvatures of velocity, temperature, viscosity and material volume fraction) and lower resolution elsewhere (Fig. 2). Our models utilise the same numerical methodologies and physical properties as those of Garel et al. (2014). A summary of the model set up is given below, and further details, including all model parameters, are provided in Supplementary Table A1.

The initial temperature field includes a cold lithosphere, with temperatures calculated using half space cooling (Turcotte and Schubert, 2002) with linear age gradient from ridge to trench both the overriding and subducting plate sides, and a prescribed bent slab down to 220 km depth. We examined both this set up with a prescribed initial slab in the upper 220 km of the box with no initial strain, as well as a set up where we kinematically drive the slab to a comparable depth before continuing the calculations without kinematic constraints. No significant differences in the evolution with the two initial conditions were observed and, accordingly, we present only the results from the former approach. We use a subducting plate with initial age





of 65 Myr and overriding plate with initial age of 20 Myr, both with the same initial length (Fig. 2a). The young overriding
120  plate exerts relatively low resistance to trench motion (Garel et al., 2014). In this study we do not vary initial subducting- and
upper-plate ages to focus on varying lithospheric and asthenospheric strengths. The top surface is a free surface, which leads
to a natural ridge push, and all other sides are free slip. Thermal boundary conditions are insulating on the left, a constant
temperature at the top and bottom, and we apply mantle temperature to the right-hand side boundary to force a ridge at the
edge of the overriding plate. The trench and the ridge on the subducting side are mobile.

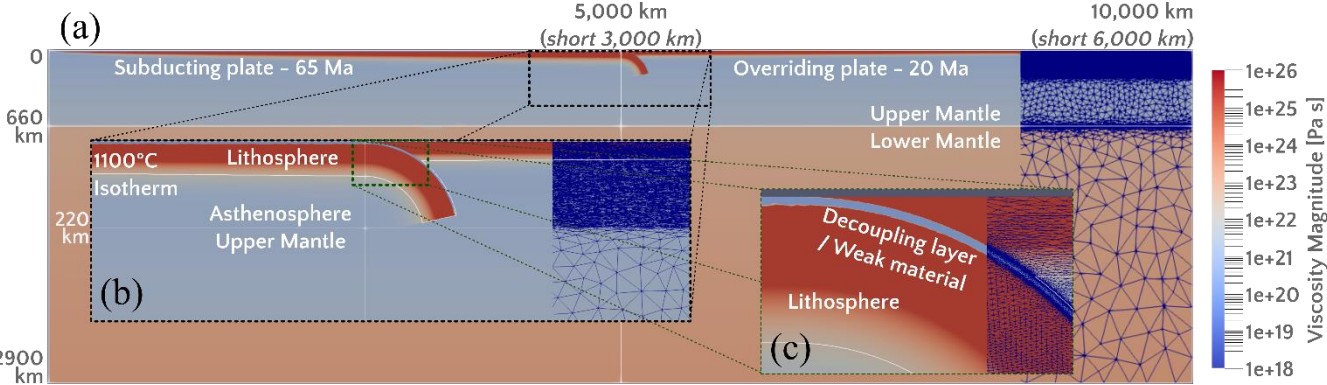

**Figure 2: Model set up illustrated with the initial viscosity field of the reference long-plate model, with text indicating parameters
for the reference short-plate model. Initial trench ages are indicated for the overriding and subducting plates, which have linear
age variations between the ridges and the trench. Initial plate temperature is set according to half space cooling (Turcotte and
Schubert, 2002). See text for boundary conditions and further information. (a) Full model view, (b) Zoom of initial slab, (c) Zoom
130  of trench and decoupling layer. The irregular adaptive mesh is presented on the right side of each panel. White lines mark the
initial location of the trench (vertical line), the ULMB at 660 km, the bottom of the asthenosphere (chosen at 220 km) and the
bottom of the lithosphere (at the 1100 °C isotherm).**

We use a temperature and strain-rate dependent composite rheology. Diffusion and dislocation creep mechanisms are
combined with a low-pressure yield-stress mechanism to approximate brittle failure and an approximation of Peierls low-
135  temperature plasticity at high pressure (e.g. Čížková et al., 2002; Garel et al., 2014; equations and parameters for the
reference cases are as in Garel et al. 2014 and are given in Appendix A). A 5-km thick weak, or decoupling, layer is included
on top of the subducting plate and slab (Fig. 2c). The decoupling layer has a relatively low viscosity, which arises from
modified rheological parameters (see Appendix A and Table A1 for further details) effective only at depths shallower than
200 km.

140  We examine the effect of basal drag by comparing long and short plates with initial length of 5000 and 3000 km from ridge
to trench, adapting both plate size, through the initial thermal field, and box size. We examine the role of asthenospheric and
lithospheric strength by adapting their rheology independently from the background rheology using a simple scaling factor.
This method allows us to isolate the influence of changing lithosphere or asthenosphere strength without some of the
additional feedbacks that arise in a fully dynamically controlled rheology. To do this, we use the 1100°C isotherm to
145  delineate the lithosphere-asthenosphere boundary (LAB) and use passive markers to track the outlines of the subducting
lithosphere and differentiate it from the overriding plate. We set the base of the asthenosphere at 220-km depth, following





the commonly imaged depth of the base of the asthenospheric low velocity zone (e.g. Dziewonski and Anderson, 1981; French et al., 2013). A 10-km linear transition is set across the boundaries of the lithosphere and the asthenosphere to prevent discontinuities in viscosity. Finally, we test the insights from these systematic models against a model where the 150 relative lithosphere/asthenosphere viscosity contrast is modified self-consistently by changing the rheological parameters.

To model a system with weak asthenosphere we multiply its viscosity by 0.5, so that it reaches minimum values of 1018-1019 Pa·s. These values are consistent with current estimates of minimum asthenospheric viscosities based on postglacial rebound and laboratory experiments (e.g. Korenaga and Karato, 2008; Paulson and Richards, 2009; Billen and Arredondo, 2018). To simulate a strong lithosphere, we multiply its viscosity by factors of 2.0 or 10.0, limiting the strongest lithosphere 155 to 1025 Pa·s (reference and most other cases) and 1026 Pa·s (strong lithosphere cases), which results in average lithospheric viscosity of 1024-1025 Pa·s. These values are consistent with the highest estimates of effective lithospheric strength (Conrad and Hager, 1999; Billen and Hirth, 2007; Burov, 2011) where 1026 Pa·s is an upper bound in terms of subductable plates. With this range of lithospheric and asthenospheric viscosities, we can generate Earth-like subduction while avoiding immediate slab detachments due to weak lithosphere resulting from high strain rates, or the stalling of subduction due to 160 unattainable forces required for the bending of very strong lithosphere.

## 3 Results

### 3.1 Reference model

The long- and short-plate reference models progress in stages, with their temporal evolution shown in Fig. 3. The initial stage is characterised by a strong slab sinking through the upper mantle at relatively high subduction velocities. 165 Asthenospheric viscosity and mantle viscosities around the slab are lowered due to high strain rates. This stage ends when the slab reaches the ULMB at 660 km depth. At this point, velocities and strain rates drop and viscosities rise.

The second stage exhibits periodic oscillations in velocity between intermediate and low values (Fig. 4a). The oscillations are caused by episodes of flattening and steepening of the slab on the ULMB and may have a surface expression of oscillating trench motion (e.g. Clark et al., 2008; Lee and King, 2011). During this stage, the slab buckles, and the resulting 170 folds are preserved during the descent of the slab into the lower mantle. The amplitude of the buckling decreases with time, such that if the models are run long enough (dependent on rheology), the subducting plates evolve to a final stage of more or less steady subduction. This is achieved in the long-plate reference model.

The shorter plate has higher subduction velocities than the long plate (Fig. 4a), whilst the longer plate exhibits enhanced trench retreat compared to the short-plate case (Fig. 4b). These results indicate a significant influence of basal drag. Higher 175 basal drag resistance leads to the slower subduction velocity of the long plate. In an additional feedback, less strain-rate weakening occurs in the asthenosphere under the long plate, as the asthenosphere moves with a similar velocity to the overlying lithosphere in each model. This enhanced asthenospheric resistance makes subduction by trench retreat





energetically preferable over subduction by plate advance over the asthenosphere. Trench advance does not occur at any stage in our reference models.

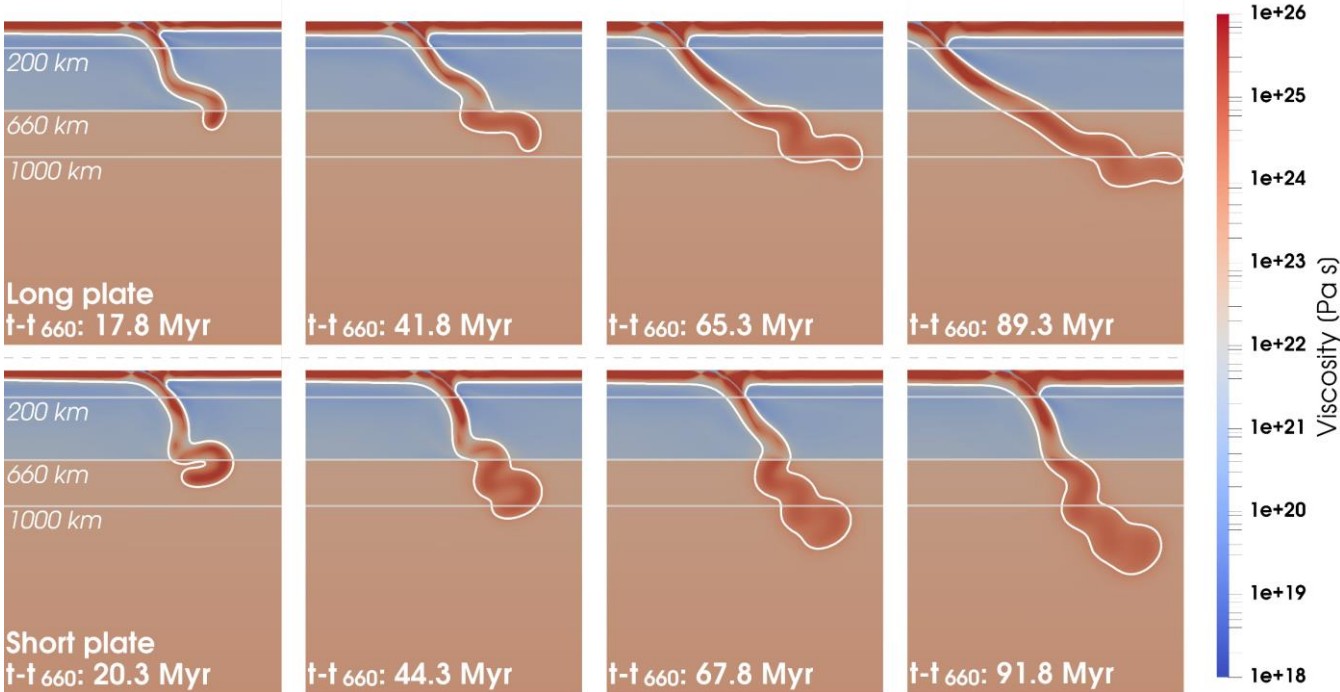

**Figure 3: Viscosity field evolution at various times for the reference long-plate model (top row) and reference short-plate model (bottom row). White contour marks the 1100 °C isotherm as the outline of the lithosphere. The vertical and horizontal spatial scales are identical and only part of the full model domain is shown. Note each model presents a distinct evolution of the slab and trench, which is the result of a difference in basal drag.**

We estimate basal drag below the subducting plate and slab pull for each time step in the models, using the following equations from Turcotte and Schubert (2002), per unit length out of the model plane:

$$F_{BD} = \frac{2 \cdot \eta_{Ast} \cdot \Delta U}{h_{Ast}} \cdot \left(2 + 3 \cdot \frac{h_{Lit}}{h_{Ast}}\right) \cdot L_{plate} \qquad (1)$$

$$F_{SP} = \Delta \rho \cdot g \cdot S_{slab} \qquad (2)$$

where $F_{BD}$ and $F_{SP}$ are the basal drag and slab pull forces respectively, $\eta_{Ast}$ is the viscosity of the asthenosphere, $\Delta U$ velocity difference between the lithosphere and the asthenosphere, $h_{Ast}$ and $h_{Lit}$ the thickness of the asthenosphere and the lithosphere, respectively, $L_{plate}$ is the length of the unsubducted part of the plate, $\Delta \rho$ is the density difference between the slab and mantle, with a typical value of 40-50 kg/m³, $g$ is the gravitational constant and $S$ is the cross-sectional area of the slab. The parameters for basal drag, $\eta_{Ast}$, $\Delta U$, $h_{Ast}$ and $h_{Lit}$, are measured at a representative vertical profile through the subducting plate: provided that the profile is taken away from the trench and ridge, there is negligible sensitivity to the profile location. For $\Delta U$, plate velocity is measured at 20 km and asthenosphere velocity at 160 km depth. The length of the slab is measured as the length of a straight line connecting the trench and the deepest point of the slab, or the coldest point of


the slab at the ULMB. Both slab pull and basal drag evolve in our dynamic models (Fig. 4d and 4e). The cyclical nature of the buckling phases is associated with the evolution of the slab pull and basal drag forces. As the slab slows and flattens on top of the ULMB as a result of the increased resistance to sinking at this depth, its dip decreases, and slab pull increases. The

higher pull increases the velocity and consequently the basal drag. The increased drag subsequently contributes to the slowing down the plate leading to steepening of the slab, and a reduction of slab pull. The following decrease in subduction velocity leads to lowering of drag. This reduction then allows for increase in slab pull and a new flattening episode.

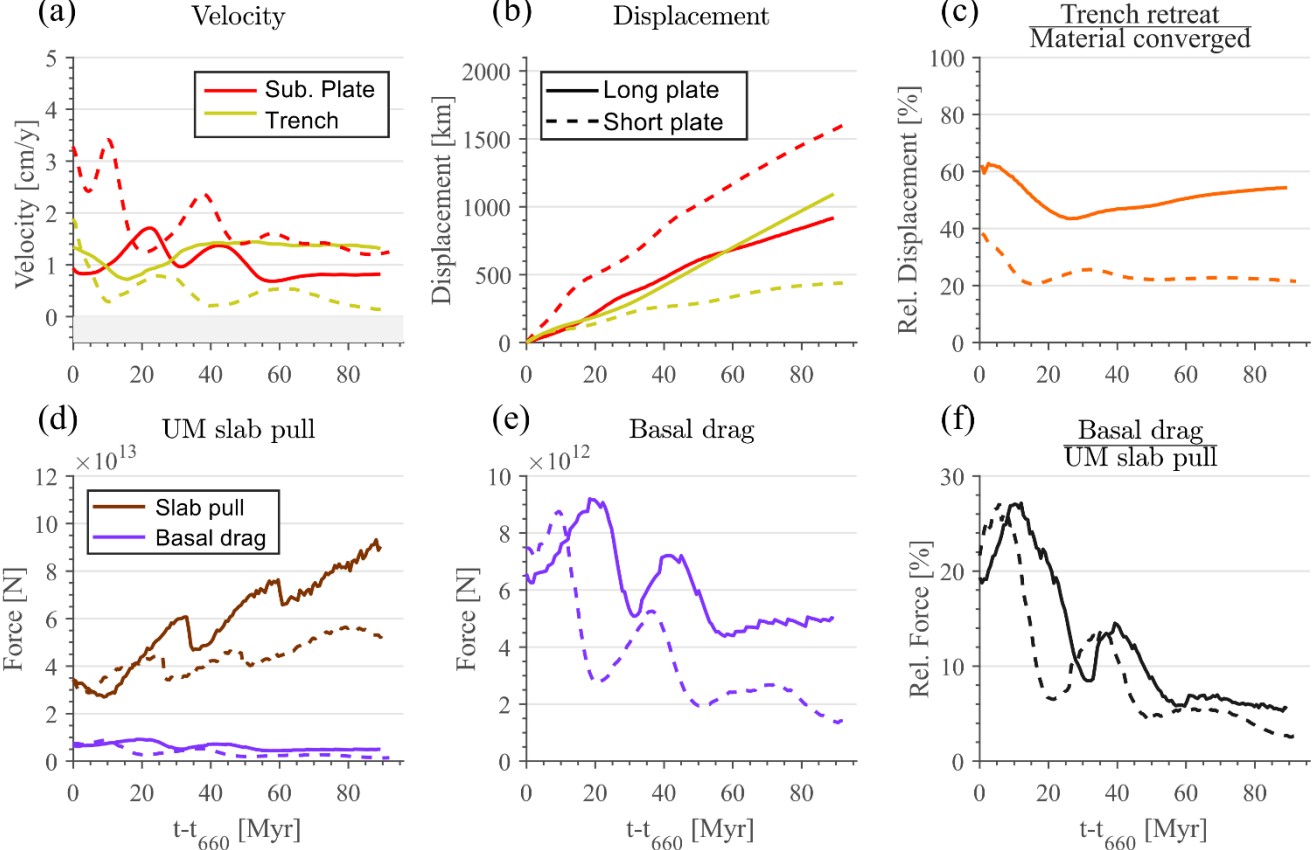

**Figure 4: Temporal evolution of the long-plate reference model (full lines) and the short-plate reference model (dashed lines). t-**

**t660 indicates the time since the initial interaction of the slab with the ULMB. (a) Velocity of the subducting plate (positive towards the upper plate, red) and the trench (positive away from the upper plate, yellow), measured at 2500 km distance from the initial subducting plate ridge. (b) Displacement of the subducting plate (red) and the trench (yellow) relative to the initial condition. (c) Percent of plate convergence (calculated as the sum of trench retreat and plate displacement) achieved by trench retreat. (d) Upper mantle slab pull and basal drag below the subducting plate, calculated as described in the main text. (e) Basal**

**drag force from (d). (f) Ratio of basal drag to upper mantle slab pull force. This shows that basal drag is of a similar magnitude as effective slab pull (i.e. the difference between gravitational pull and the viscous mantle resistance on the slab sides) which is ~10% of the upper-mantle slab pull force.**

For the reference models, we find that basal drag averages around 10% of slab pull, ranging from just below 30% in the fast early stage of subduction to 3-5% in the final stage (Fig. 4f). This is similar to the estimated magnitude of ridge push relative

to slab pull (e.g. Turcotte and Schubert, 2002). Furthermore, the basal drag ratio (of slab pull) for the short plate is





systematically several % lower than the basal drag ratio for the long plate (Fig. 4e and 4f). This difference in basal drag ratio between the long- and short-plate models is apparently sufficient to change subduction velocity and influence the overall evolution of the slab's morphology.

## 3.2 Weak asthenosphere

The most direct mechanism to reduce basal drag is to lower the asthenosphere viscosity. A lower basal drag allows for higher subduction velocity to develop, which in turn decreases the asthenospheric viscosity through the feedback effect of increased strain rate. The compound effect of applying the 0.5 reduction factor and consequent strain-rate weakening is an order of magnitude decrease in asthenospheric viscosity compared to the reference models.

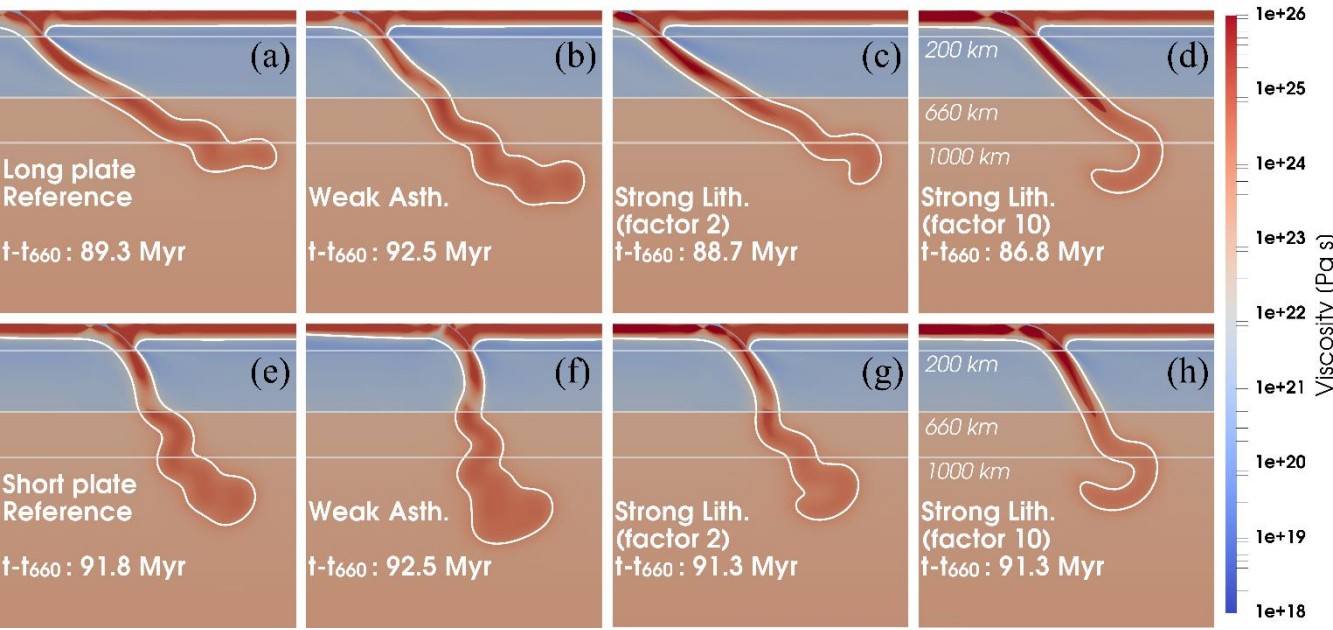

**Figure 5: Comparison of final slab shapes for different models shown as a snapshot of the viscosity field. Long-plate (top row, (a)–(d)) and short-plate (bottom row, (e)–(h)) cases of Reference models ((a) and (e)), Weak Asthenosphere models ((b) and (f)), and Strong Lithosphere models with strength scaling factors of 2 ((c) and (g)) and 10 ((d) and (h)). White contour marks the 1100 °C isotherm as the outline of the lithosphere. The vertical and horizontal spatial scales are identical, with grey lines to mark depths of 200 km, 660 km and 1000 km. For every type of model, the subducting slab in the short-plate model has more buckles and has**
**undergone less trench retreat than in the long-plate model.**

Assigning a weak asthenosphere results in an increase in subduction velocity and a reduction in trench retreat, as shown in Fig. 5 and Fig. 6. However, the reduced basal drag does not remove the plate-length dependence of the subduction evolution; the difference between the displacement of the subducting plate for the long and short plates remains similar even if the total displacement has slightly increased due to the higher subduction velocities (Fig. 6a). The basal-drag ratios are lower than in
the reference models and the difference between the long and short plates has increased, with the basal drag ratio for the short plate ranging from about 20% to less than 1% over the evolution of the models (Fig. 6d). Although overall ratios of basal drag to slab pull are lower than in the reference model, basal drag still plays significant role in these low-viscosity



asthenosphere cases, as is clear from the difference in the final shapes of the long- and short-plate slabs (Fig. 5b and 5f). The overall contribution of trench retreat to plate convergence is reduced for both the long- and the short-plate models compared

with the reference models (Fig. 6c).

### 3.3 Strong lithosphere effect (factor 2 and 10)

A second mechanism to reduce the significance of basal drag as resistive force is to increase the energy cost of bending through increased lithospheric viscosity (Conrad and Hager, 1999). This was implemented by applying the strengthening factor only in the subducting lithosphere and the slab. This process, unlike the reduction in asthenosphere viscosity, does not

trigger any strong feedbacks between strain rate and effective viscosity. We therefore tested both factor 2 and factor 10 to compare the strengthening mechanism with the prescribed and the effective asthenosphere weakening. The resulting plate strength of the factor 10 experiments was at the upper bound of suggested realistic plate viscosity and at the upper bound of what is subductable in our models.

Increasing the strength of the lithosphere by a factor 2 reduces the amount of material subducted for the short-plate model,

relative to the short-plate reference model, much more than it reduces the amount subducted for the long plate compared to the long-plate reference case (i.e. the behaviour of the long and the short plates becomes more similar). Trench retreat for both plates is similar to the reference models. The resulting basal drag ratio is slightly reduced relative to the reference models (by 1-2%), as shown in Fig. 6e-6h. Slab shapes at the end of the simulation are similar to the reference models with somewhat less slab deformation and buckling at the ULMB interface because less material has subducted (Fig. 5c).

For the factor 10 increase of subducting-lithosphere viscosity, the long- and short-plate models achieve similar amounts of subducted material. The amount of material subducted by the long plate deviates substantially less from the reference than the short plate, while trench retreat is reduced more in the long- than the short-plate model (Fig. 6i and 6j). This is as expected following the work of Conrad and Hager (1999); the resistance to plate bending becomes a more important control on subduction velocity than basal drag if the plate is sufficiently strong. However, we consider this model as an end-member

of plate strength, as it is close to what, in our models, is the limit of subductable plates. The basal drag ratio for these strong-plate models is substantially lower than the reference models, with average values of around 5% only, and with a small systematic difference remaining between the long- and short-plate models (Fig. 6l). This amount of reduction in basal drag ratio by about 3-5% is sufficient to influence subduction dynamics, as can be observed at the final stages of the models (Fig. 5d) where the long- and short-plate models achieve different trench locations and slab shapes than the reference models.

In models where we combined a weaker asthenosphere with a stronger lithosphere (factor 2 and factor 10), the result was a joint effect of what we observed when we changed asthenosphere and lithospheric viscosity individually. The combined models subduct an increased amount of plate but there was less difference between the long- and short-plate models, and trench retreat was reduced. Thus, the models with strong lithosphere and weak asthenosphere exhibit the most Earth-like behaviour in terms of a low contribution of trench motion to convergence and limited sensitivity to plate length. However,

this behaviour only appears for the upper bound of plate strength.





**Figure 6: Temporal evolution of the long-plate (full lines) and short-plate (dashed lines) models. Rows of graphs are for different types of models, where (a)–(d) are Weak Asthenosphere models, (e)–(h) are Strong Lithosphere (factor 2) models and (i)–(l) are Strong Lithosphere (factor 10) models. The first column, (a), (e) and (i), shows the displacement of the subducting plate from the initial state, the second column, (b), (f), and (j), the amount of trench retreat, the third column, (c), (g) and (k), shows trench retreat significance as percentage of the total material converged, the fourth column, (d), (h) and (l), shows the ratio of basal drag to upper mantle slab pull force. All models are compared with the reference models, shown in light red.**





### 3.4 Self-consistent rheology

We also tested whether a self-consistent rheology that yields a stronger lithosphere and weaker asthenosphere than the
reference models behaves similarly as the models where we artificially prescribed regions of modified viscosity. To do this,
we used a simple optimisation algorithm to find a combination of flow law parameters that yielded an increase/reduction in
viscosity in the lithosphere/asthenosphere (further detail in Maunder et al., 2016). We required that the modified composite
rheology still satisfied the set of Earth-like conditions that Garel et al. (2014) imposed. These conditions include: (i) that
viscosities of the upper mantle and lower mantle will be in a range reasonable for postglacial rebound (Paulson and Richards,
2009), (ii) Peierls low temperature plasticity is required to control deformation in the lower lithosphere and in the slab once
subducted, to account for deep slab weakening, (iii) the transition from dislocation to diffusion creep as the dominant
deformation mechanism is required to comply with the depth range inferred from seismic anisotropy (i.e. around 250 km
depth in background mantle (Ranalli, 1995)). We optimised the values of the activation energy and volume, the pre-factor
and the stress exponent values of diffusion, dislocation and Peierls creep to achieve a lower asthenospheric and higher
lithospheric viscosity without significantly modifying the lithospheric thickness of the reference models. Supplementary
Table A1 lists the set of modified parameters that satisfies all constraints. We then tested that under the new rheology,
subduction remained viable. We found that, in addition to adjusting the overall rheological parameters, we needed to
increase the strength of the weak layer to limit subduction velocities (and avoid slab detachment) early in the evolution of the
model.
All these conditions together strongly narrow the set of possible rheological parameters that allow strengthening of the
lithosphere while weakening the asthenosphere, and we were not able to strengthen the lithosphere as much as in the scaled
rheology models without stalling subduction. This is due to the nature of self-consistent rheology, which couples the strength
of the plate and its thickness. Any plates with 10 times higher strength were also thicker thus additionally increasing bending
resistance and impeding subduction. Supplementary Fig. B2 shows how the evolution of subducting plate morphology and
trench retreat are similar to our models where we artificially lower the viscosity of the asthenosphere and increased the
viscosity of the lithosphere.

### 3.5 Models with extended domain

In the models discussed above, we changed the size of the model domain together with plate size, so that the models always
contained the only two plates with their bounding spreading ridges at the box edge. To test whether box size affects the
results, we also ran a set of models where we use an extended and fixed box size but vary the plate size inside the box. We
tested both the long- and the short-plate reference cases in a box of 15,000 km length. Compared to the reference models, the
box was extended by an additional 5000/9000 km, for the long- and short-plate models respectively, to the left (trailing end)
of the subducting plate. The initial condition in the extended section was set to mantle temperature throughout, i.e. the
extended section contained no initial lithosphere. The upper plate was set to be the same length as the reference models (i.e.





5000 km and 3000 km for the long plate and short plate models respectively). The top thermal boundary condition allowed

for lithosphere to develop in the extended section with time. Comparing with the reference models, these cases display the

same trends of plate advance and trench motion, with differences in diagnostic parameters of less than 5% (Fig. 7). These

differences are minor and indicate that box size does not bias the results.

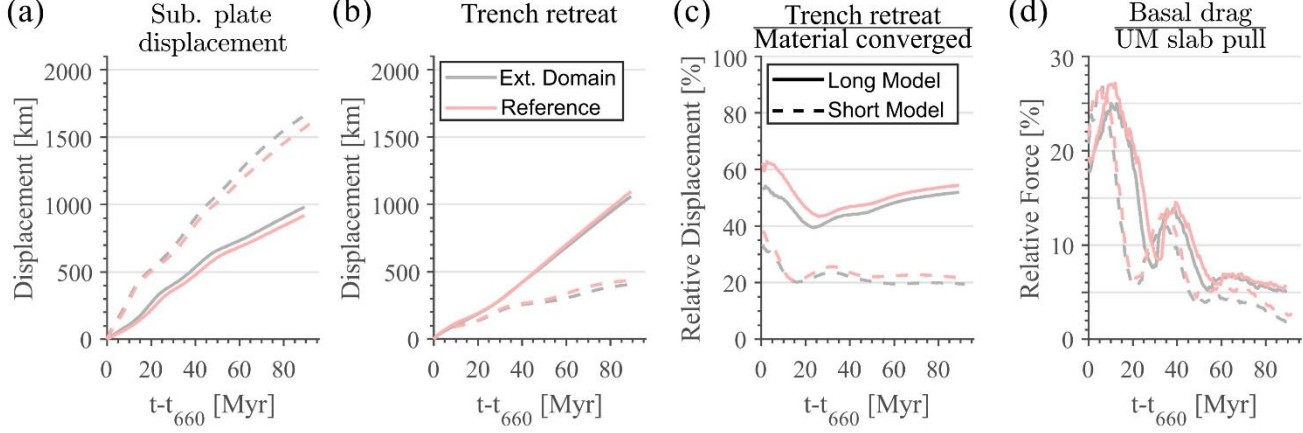

**Figure 7: Temporal evolution of the long-plate (full lines) and short-plate (dashed lines) extended domain (grey) and reference (light red) models. (a) displacement of the subducting plate from the initial state. (b) amount of trench retreat. (c) trench retreat significance as percentage of the total material converged, (d) ratio of basal drag to upper mantle slab pull force.**

## 4 Discussion

### 4.1 Trench motion

The trench always retreats in both the long- and short-plate reference models. Although strengthening the lithosphere and

weakening the asthenosphere affect trench behaviour, only the short-plate model with the weak asthenosphere displays

periods of trench advance. In this model, due to the lowered basal drag, subduction is able to occur through more plate

advance, whilst limited periods of trench advance occur during the buckling interaction between the strong slab and lower

mantle. These periods occur when increased subduction velocity brings colder and stronger slab to interact with the ULMB.

The interaction between the steep strong slab and the upper mantle, which has lowered viscosity due to strain-rate

weakening, does not allow the slab to unbend. Instead, the cold slab attains a rolled-over shape that upon interaction with the

higher viscosity lower mantle pushes the plate forward. These episodes end once subduction velocity reduces, as this gives

the slab time to heat and weaken, and to regain the ability to unbend. This type of trench advance mechanism can be

observed in other models (Bellahsen et al., 2005; Di Giuseppe et al., 2009; Ribe, 2010; Billen and Arredondo, 2018), in

particular models where slab strength does not decrease with increasing temperature (i.e. models without a Peierls type

plasticity). In nature, only below part of the Indian collision front has a slab with such a rolled-over shape been imaged (e.g.

review by Goes et al., 2017); below other advancing trenches, slabs are steep but do not have the shape expected for trench

advance resulting from the push of a strong slab at the ULMB. So, although the modelled trench advance episodes have





magnitudes and rates in the range of observations, other mechanisms for trench advance, such as external forcing by
background mantle flow and mantle plumes (as happens in the models from Höink and Lenardic, 2008) or an imbalance
between ridge push and slab pull (e.g. Capitanio et al., 2009) may be more likely drivers of trench advance on Earth.

**4.2 Model limitations**

Other approximations made in the models are that we study only a single initial age for subducting and overriding plates and
neglect the effect of upper mantle phase transitions. Phase transitions exert a second-order effect on the difference between
short- and long-plates, that would mainly enhance the slab buckling and flattening that happens in the models (see e.g. Goes
et al., 2017 for a review). The effect of plate age was studied by Garel et al. (2014) and Agrusta et al. (2017), who used a
similar model set up as we use here but varied the age of subducting and upper plates without varying plate size. These
models show that the initial ages of subducting and overriding plates affect the partitioning of plate convergence between
trench retreat and subducting plate advance, with a lower contribution of trench retreat for initially younger subducting plates
and initially older overriding plates, resulting in a steeper more buckled slab. The difference in motion partitioning and
resulting slab morphology between our short- and long-plate reference models is similar to the difference between
subduction of an initially 100 Myr old and an initially 30-50 Myr old plate (Garel et al., 2014). This further emphasises the
importance of drag below the plate in the total force balance of our models.

However, one should also bear in mind the limitations of the 2D setup of the models. For 3D plates, there may be more
convergent boundary length (and hence slab pull) relative to plate surface area than in the 2D cases. This influences the ratio
of slab pull to basal drag, which we will investigate for the main natural subduction systems in the following section.
Furthermore, in a 3D system mantle flow may be in an angle to plate advance direction, which will affect the component of
velocity used to calculate the drag. A more Earth-like 3D system will also include mantle upwelling as well as downwelling
flows. Upwelling flow, together with the global system of subduction-driven flow can result in driving drag below some
plates (e.g. Colli et al., 2014; Stotz et al., 2018; Coltice et al., 2019), and cannot occur in our model setup. These
complications, as well as local phenomena (e.g. the presence of mantle plumes, lateral variations in slab properties, an
imbalance between ridge push and slab pull (Capitanio et al., 2009; Capitanio, 2013)) may result in more complex behaviour
than presented in our models.

**4.3 The role of basal drag on Earth**

Our set of models clearly shows that basal drag is an important component of the force balance governing subduction
dynamics, across the parameter space investigated. This is consistent with simple analytical calculations of basal drag, as we
next demonstrate, by calculating basal drag and slab pull for the Pacific and Cocos plate as large and small plate end
members. We use Eq. (1) and (2) but for plate area $S_{plate}$ rather than length $L_{plate}$ and slab volume $V_{slab} = S_{slab} \cdot L_{trench}$
rather than slab area $S_{slab}$. We assume only the upper mantle part of the slabs pulls and slabs extend through the whole upper
mantle under an average dip of 80° (Lallemand et al., 2005). The properties used for the calculation and the forces calculated





are summarised in Table 1. The relative basal drag, measured as the ratio of basal drag to slab pull, is around 16% for the Pacific plate, but only 5% for the Cocos Plate. The analytical values of basal drag are somewhat larger than the values in the second stage of our models because the Pacific plate is about a factor of two larger than our subducting plate in the long-plate model, and our model velocities are on the low side compared to those observed around the Pacific. So, all evidence

points to a role of basal drag that is significant, and the models would predict that unless other factors mask this trend, subduction velocities should vary with plate size.

| | $U$ $\left[\dfrac{cm}{year}\right]$ | $S$ $[km^2]$ | $L_{trench}$ $[km]$ | $\eta_{Ast}$ $[Pa \cdot s]$ | $\Delta\rho$ $\left[\dfrac{kg}{m^3}\right]$ | $h_{Lit}, h_{Ast}$ $[km]$ | $W_{Slab}$ $[km]$ | $F_{SP}$ $[N]$ | $F_{BD}$ $[N]$ |
|---|---|---|---|---|---|---|---|---|---|
| Pacific | 6.3 | $108 \cdot 10^6$ | $12 \cdot 10^3$ | $10^{19}$ | 50 | 100,150 | 80 | $3.5 \cdot 10^{19}$ | $0.16 \cdot 10^{19}$ |
| Cocos | 6.6 | $3 \cdot 10^6$ | $2.5 \cdot 10^3$ | | | | 40 | $33.6 \cdot 10^{19}$ | $5.52 \cdot 10^{19}$ |

**Table 1: Slab pull and basal drag force calculations, with associated parameters, for the Pacific and Cocos plates. $U$ is plate subduction velocity, $S$ is plate surface area, $L_{trench}$ is the length of subduction trench, $\eta_{Ast}$ is the viscosity of the asthenosphere, $\Delta\rho$ is the density difference between the slab and surrounding mantle and $h_{Lit}$ and $h_{Ast}$ are the thickness of the lithosphere and**
**asthenosphere. $W_{Slab}$ is the thickness of the slab, contributing to the difference in buoyancy due to slab age. $F_{SP}$ and $F_{BD}$ are the slab pull and basal drag forces, calculated using Eq. (1) and (2).**

We re-examine the observations of velocity and plate size throughout the Cenozoic Era, using GPlates (Müller et al., 2018) and the new reconstruction of Müller et al. (2016). We include only plates mainly driven by their attached subducting slabs, consistent with Conrad and Hager (1999). The trenches considered are those where subduction occurs below North, Central

and South America, Alaska-Aleutians, Kuriles-Japan, Izu-Bonin-Marianas, Ryukyu-Philipinnes, Tonga-Kermadec and Sumatra-Java, with the Pacific, Philippine, Cocos, Nazca, Juan de Fuca, Farallon, Indo-Australia, Izanagi, and Kula plates as subducting plates. At 10 Myr intervals between 0 and 60 Myr ago, we evaluate which of these subduction systems are active. Maps of the trenches considered at each time are included in Fig. B1. In our velocity and age estimates, we remove points at the trench ends to avoid biases by edge effects (e.g. highly oblique motions – see details in Appendix C). We average age

and velocity of the active subduction systems along each trench at each time step and calculate plate area for the plate polygons that make up the subducting plate at the corresponding time. Resulting trends are similar for the full velocity of the subducting plate and the trench-normal component of it.

The resulting size-velocity trend is shown in Figure 8A. The newer reconstruction shows a similar present-day trend (bold symbols) as the older reconstructions (Fig. 1 and Fig. C2), with a possible trend of increasing subduction velocity with

increasing plate size for small plates, but a flat trend for plates with a larger than $3000^2$ km². When all subducting plates through the Cenozoic are considered, no significant trend of velocity with age remains. However, what is seen (Fig. 8b) is that, on average, subduction zones with older plate at the trench tend to be bigger, like the present-day Pacific and Indo/Australian plate, than those subducting younger material at the trench, such as the present-day Cocos, Nazca and Juan de Fuca.





Previous (thermo-)mechanical models show that the age of the subducting plate strongly affects subduction dynamics. Older plates, i.e. those with high age at the trench, are colder and therefore denser and stronger. Consequently, they exert more pull force and are harder to bend, resulting in higher subduction velocities and higher trench retreat rates (Bellahsen et al., 2005; Capitanio et al., 2007; Stegman et al., 2010; Garel et al., 2014). A consequence of the observed correlation of average age at the trench and plate size is that plates with a stronger slab pull tend to also have a stronger resisting basal drag. This could (at

least in part) explain the observation that velocities of subducting plates on Earth, today and throughout the Cenozoic, tend to be mostly fall around 8-10 cm/yr (Fig. 8, Sdrolias and Müller, 2006; Goes et al., 2011).

Note that the relation between age and size is not a causal relationship, but a feature of the plate configuration that has dominated most of the Cenozoic. Early in the Cenozoic, there are several cases that deviate from the buffered velocity trend. At the start of the Cenozoic, much of the subduction surrounding the Pacific plate consumed relatively young lithosphere,

even though the Pacific plate itself was already large in size (horizontally aligned points at the top of the area–age trend in Fig. 8b). Other early Cenozoic deviations include very high velocities of the last remnants of the Izanagi and Kula plates (points with area of about $2500^2$ km$^2$ and velocities around 17 cm/yr in Fig. 8a) and low velocities of Farallon plate (area about $6000^2$ km$^2$ in Fig. 8a).

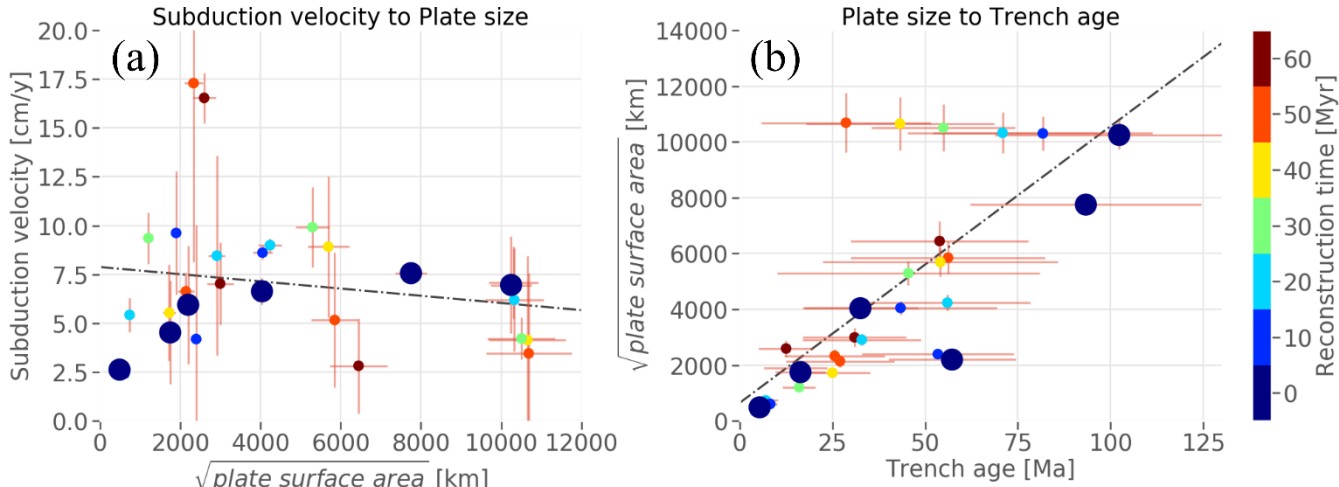

**Figure 8: (a) Velocity of subducting plates from Müller et al. (2016) in lower mantle (plume) reference frame (O'Neill et al., 2005) throughout the Cenozoic, as a function of characteristic plate size (measured as the square root of plate area). (b) Plate size as a function of age at the trench for subducting plates throughout the Cenozoic. Maps showing the subduction zones included at each time can be found in Fig. C1. Blue bold circles represent data from present-day subducting plates. Linear regression fits (black lines) show that there is no resolvable trend of velocity with plate size (p-value 0.351, R$^2$ 0.03) while there is a trend of size with age**

**(p value < 0.05 and R$^2$ of 0.52).**

Previous models documented that older plates tend to drive more trench retreat due to both their larger bending strength and their larger negative buoyancy (e.g., Capitanio et al., 2007; Stegman et al., 2010; Garel et al., 2014). This has been suggested to help explain the difference between Cenozoic subduction styles along the western Pacific (subducting mainly older plate with significant trench retreat) and eastern Pacific (subducting younger plates with low trench retreat contributions) (Goes et



al., 2017). Our results suggest that plate size may further enhance these differences, as the high basal drag exerted on the dominantly westward-subducting large Pacific plate would favour more trench retreat than for the smaller eastward-subducting Cocos, Nazca and Juan de Fuca plates.

**5 Conclusions**

We have presented a set of 2D models to examine the role of basal drag in subduction dynamics. Previous studies have reached contradictory conclusions, with some proposing that drag between the subducting plate and the underlying asthenosphere plays only a small role in dictating the subduction force balance (e.g. Conrad and Hager, 1999; Billen and Arredondo, 2018; Wolf and Huismans, 2019), but others invoking a more significant role for basal drag in controlling subduction and plate motions on Earth (e.g. Conrad and Lithgow-Bertelloni, 2006). A low contribution of basal drag could explain the lack of a correlation between plate velocity and size (Conrad and Hager, 1999). Furthermore, the observation that trench migration velocities are mostly low compared to convergence velocities, can be either positive or negative in sign (Billen and Arredondo, 2018), and may be the result of a high ratio of lithosphere to asthenosphere viscosity drag (Conrad and Hager, 1999; Höink and Lenardic, 2008; Billen and Arredondo, 2018), is also suggestive of a low contribution from basal drag.

Our 2D numerical models comprise a single subduction system with a mobile trench and composite non-linear rheology. We consider long and short subducting plate models to evaluate the effect of plate length and the associated basal drag. The set-up of these regional subduction models is similar to those of previous studies, which have been used to elucidate how slab buoyancy, slab strength, slab interaction with the upper-lower mantle boundary or subduction-upper plate interaction influence subduction dynamics (e.g. Bellahsen et al., 2005; Capitanio et al., 2007, 2009; Stegman et al., 2010; Čížková and Bina, 2013; Garel et al., 2014; Agrusta et al., 2017; Billen and Arredondo, 2018; Wolf and Huismans, 2019). We found that basal drag limits subduction velocities for our reference models and models with a lower asthenospheric viscosity (in a range consistent with likely viscosities on Earth and still allowing for steady subduction, without rapid slab detachment). Models with a low-viscosity asthenosphere do reduce the contribution of trench motion to plate convergence to more Earth-like values, as observed in previous studies (e.g. Capitanio et al., 2007; Billen and Arredondo, 2018). They also allow for periods of trench retreat and advance (Billen and Arredondo, 2018), although we note that trench advance with reclining/vertical slab geometries as observed in nature probably requires additional forcing (e.g. by upwellings or other forces within the global plate system).

Strengthening the lithosphere can significantly lower the dependency of velocity on plate length but requires strengthening by as much as a factor of 10, which is an upper bound: our models that incorporate lithospheric strengthening of this magnitude often preclude subduction. Although it has been shown that the relative strength of the lithosphere to asthenosphere is important in controlling the dynamics of subduction (Höink and Lenardic, 2008, 2010; Ribe, 2010), with non-linear rheologies, increasing lithosphere strength and decreasing asthenosphere strength have distinct effects.





Basal drag values in our models range from a few percent of slab pull (in the strongest lithosphere cases) to up to 10-30%. Similar values are estimated in analytical calculations of basal drag to pull ratio assuming a reasonable set of parameters for the Pacific and Cocos plate. This, together with results from global flow models where basal drag by whole mantle flow has

been shown to be important to reproduce patterns of plate motion (Lithgow-Bertelloni and Richards, 1998; Becker and O'Connell, 2001; Conrad and Lithgow-Bertelloni, 2006), indicates that basal drag is a substantial contributor towards the subduction force balance, of the same order as effective slab pull (i.e. slab pull minus the viscous resistance to sinking) and ridge push.

Based upon an analysis of Cenozic plate motion reconstructions, we suggest that the reason that most plates move at

velocities around 8-10 cm/yr is because plate size correlates with plate age at the trench (i.e., both driving and resisting forces increase together). As a result, the increase in basal drag more or less balances the increase in plate velocity induced by increased slab pull with increasing age. This co-dependency between plate velocity, age and length should be considered in regional models of subduction systems.

**Appendix A, governing equations and rheology calculations in the models**

We solve flow for incompressible Stokes fluid, under the Boussinesq approximation, assuming mass, momentum and energy conservation equations:

$$\partial_i u_i = 0 \tag{A1}$$

$$\partial_i \sigma_{ij} + \Delta\rho g_j = 0 \tag{A2}$$

$$\frac{\partial T}{\partial t} + u_i \partial_i T - \kappa \partial_i^2 T = 0 \tag{A3}$$

where $u$ is the velocity, $\sigma$ is the stress tensor, $g$ is gravity, $T$ is temperature, $\kappa$ is the thermal diffusivity and $\Delta\rho = -\alpha\rho_s\Delta T$ is the density difference due to temperature, with $\alpha$ the coefficient of thermal expansion, $\rho_s$ the reference (surface) mantle density and $\Delta T$ the difference in temperature from the surface.

Viscosity is therefore the ratio of deviatoric stress to strain rate:

$$\mu = \frac{\tau_{ij}}{2\dot{\varepsilon}_{ij}} = \frac{\sigma_{ij} + P\delta_{ij}}{2\dot{\varepsilon}_{ij}} \tag{A4}$$

where $\tau$ is the deviatoric stress, $\dot{\varepsilon}$ is the strain rate, $P$ is the dynamic pressure and $\delta_{ij}$ is the delta function. The viscosity is calculated in our models using:

$$\mu = \left( \frac{1}{\mu_{diff}} + \frac{1}{\mu_{disl}} + \frac{1}{\mu_y} + \frac{1}{\mu_{Pie}} \right)^{-1} \tag{A5}$$



where $\mu_{diff}$, $\mu_{disl}$, $\mu_y$ and $\mu_{Pie}$ are the viscosities calculated using diffusion creep, dislocation creep, yielding mechanism and simplified Pierels creep, respectively. The viscosities derived from diffusion, dislocation and Pierels creep are calculated

using the generalised equation:

$$\mu_{diff \backslash disl \backslash Pie} = A^{\frac{-1}{n}} \exp\left(\frac{E + P_l V}{nRT_{ad}}\right) \dot{\varepsilon}_{II}^{\frac{1-n}{n}} \tag{A6}$$

$$P_l = \rho g D \tag{A7}$$

where $A$ is a prefactor, $n$ is the stress exponent, $E$ and $V$ are the activation energy and volume, respectively, $P_l$ is the lithostatic pressure, $R$ the gas constant, $\dot{\varepsilon}_{II}$ is the second invariant of the strain rate tensor and $D$ is the depth. $T_{ad}$ is the

temperature adjusted with an adiabatic gradient of 0.5°K/km in the upper mantle and 0.3°K/km in the lower mantle (Fowler, 2005). The yielding mechanism is calculated as:

$$\mu_y = \frac{\tau_y}{2\dot{\varepsilon}_{II}} = \frac{\min(\tau_s + f_c P_l, \tau_{y\,max})}{2\dot{\varepsilon}_{II}} \tag{A8}$$

where $\tau_y$ is the yield stress, $\tau_s$ is the surface yield stress, $f_c$ is the friction coefficient and $\tau_{y\,max}$ is the maximum yield stress. The viscosity field is capped by both minimum and maximum values. The yielding viscosity is adjusted within the weak

decoupling layer by applying a different friction coefficient:

$$\mu_{y\,weak} = \frac{\min(\tau_s + f_{c\,weak} P_l, \tau_{y\,max})}{2\dot{\varepsilon}_{II}} \tag{A9}$$

The initial temperature field in the lithosphere is calculated using the half-space cooling equation from Turcotte and Schubert (2002):

$$T = T_s + \Delta T \cdot \text{erf}\left(\frac{D}{2\sqrt{\kappa t_{age}}}\right) \tag{A10}$$

where $T_s$ is the surface temperature and $t_{age}$ is the age.

| Quantity | Symbol | Units | Value | | |
|---|---|---|---|---|---|
| | | | UM (Reference) | UM (Self-Consistent) | LM |
| Gravity | $g$ | $m \cdot s^{-2}$ | 9.8 | | |
| Thermal expansivity coefficient | $\alpha$ | $K^{-1}$ | $3.0 \cdot 10^{-5}$ | | |
| Thermal diffusivity | $\kappa$ | $m^2 \cdot s^{-1}$ | $10^{-6}$ | | |
| Reference (surface) density | $\rho_s$ | $kg \cdot m^{-3}$ | 3300.0 | | |
| Cold, surface temperature | $T_s$ | $K$ | 273.0 | | |
| Hot, mantle temperature | $T_m$ | | 1573.0 | | |





| | | | | | |
|---|---|---|---|---|---|
| Gas constant | $R$ | $J \cdot K^{-1} \cdot mol^{-1}$ | 8.3145 | | |
| Maximum viscosity (Strong Lithosphere model) | $\mu_{max}$ | $Pa \cdot s$ | $10^{26}$ | | |
| Maximum viscosity (Reference and all other models) | | | $10^{25}$ | | |
| Minimum viscosity | $\mu_{min}$ | | $10^{18}$ | | |
| **Diffusion Creep** | | | | | |
| Activation energy | $E$ | $J \cdot mol^{-1}$ | $300.0 \cdot 10^3$ | $335.0 \cdot 10^3$ | $200.0 \cdot 10^3$ |
| Activation volume | $V$ | $m^3 \cdot mol^{-1}$ | $4.0 \cdot 10^{-6}$ | $5.0 \cdot 10^{-6}$ | $1.5 \cdot 10^{-6}$ |
| Pre-factor | $A$ | $Pa^{-n} \cdot s^{-1}$ | $3.0 \cdot 10^{-11}$ | $1.5 \cdot 10^{-9}$ | $6.0 \cdot 10^{-17}$ |
| Stress exponent | $n$ | | 1.0 | | |
| **Dislocation Creep (UM)** | | | | | |
| Activation energy | $E$ | $J \cdot mol^{-1}$ | $540.0 \cdot 10^3$ | $472.0 \cdot 10^3$ | $300.0 \cdot 10^3$ |
| Activation volume | $V$ | $m^3 \cdot mol^{-1}$ | $12.0 \cdot 10^{-6}$ | $11.0 \cdot 10^{-6}$ | $2.0 \cdot 10^{-6}$ |
| Pre-factor | $A$ | $Pa^{-n} \cdot s^{-1}$ | $5.0 \cdot 10^{-16}$ | $1.34 \cdot 10^{-17}$ | $10^{-42}$ |
| Stress exponent | $n$ | | 3.5 | 3.472 | 3.5 |
| **Peierls Creep (UM)** | | | | | |
| Activation energy | $E$ | $J \cdot mol^{-1}$ | $540.0 \cdot 10^3$ | $540.0 \cdot 10^3$ | $300.0 \cdot 10^3$ |
| Activation volume | $V$ | $m^3 \cdot mol^{-1}$ | $10.0 \cdot 10^{-6}$ | $10.0 \cdot 10^{-6}$ | $2.0 \cdot 10^{-6}$ |
| Pre-factor | $A$ | $Pa^{-n} \cdot s^{-1}$ | $10^{-150}$ | $10^{-145}$ | $10^{-300}$ |
| Stress exponent | $n$ | | 20.0 | | |
| **Yield Strength Law** | | | | | |
| Surface yield strength | $\tau_s$ | $MPa$ | 2.0 | | |
| Friction coefficient | $f_c$ | | 0.2 | | |
| Friction coefficient (decoupling layer) | $f_{c\,weak}$ | | 0.02 | 0.02 | 0.02 |
| Maximum yield strength | $\tau_{y\,max}$ | $MPa$ | 10,000 | | |

**Table A1: Physical and rheological parameters of all models.**





**Appendix B, illustration of the evolution of all models discussed in the main text**

## Reference

## Weak asthenosphere

**Figure B1: Evolution of all types of models (top – Long models, bottom – Short models)**



## Strong lithosphere (factor 2)

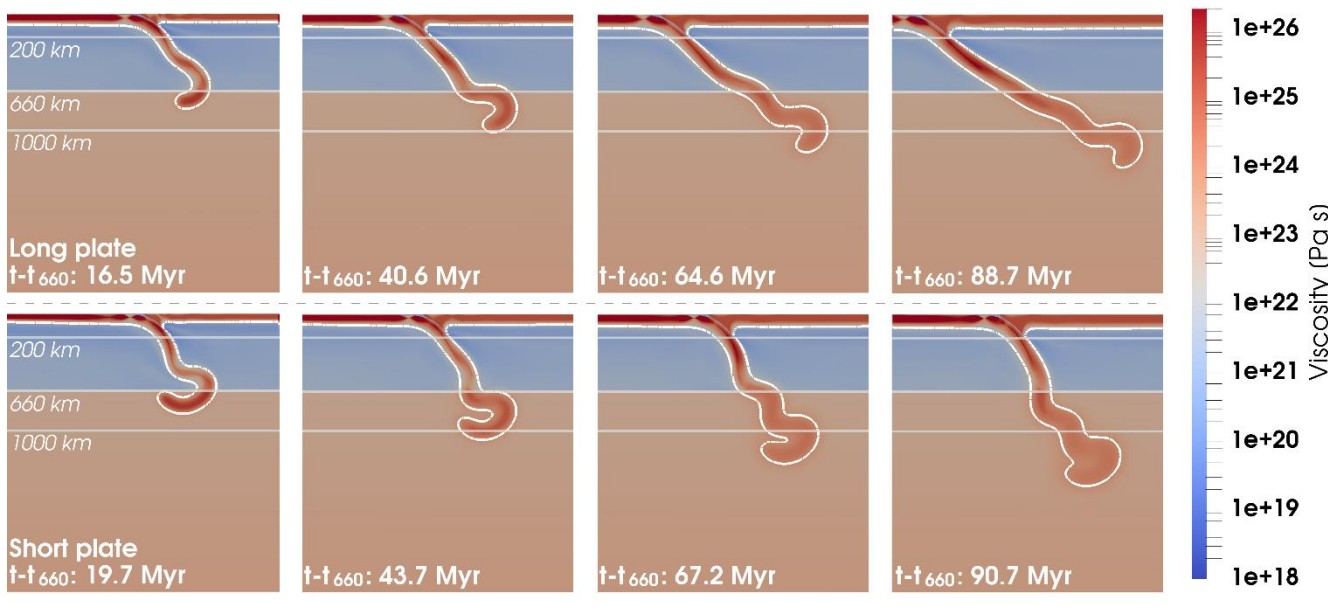

## Strong lithosphere (factor 10)

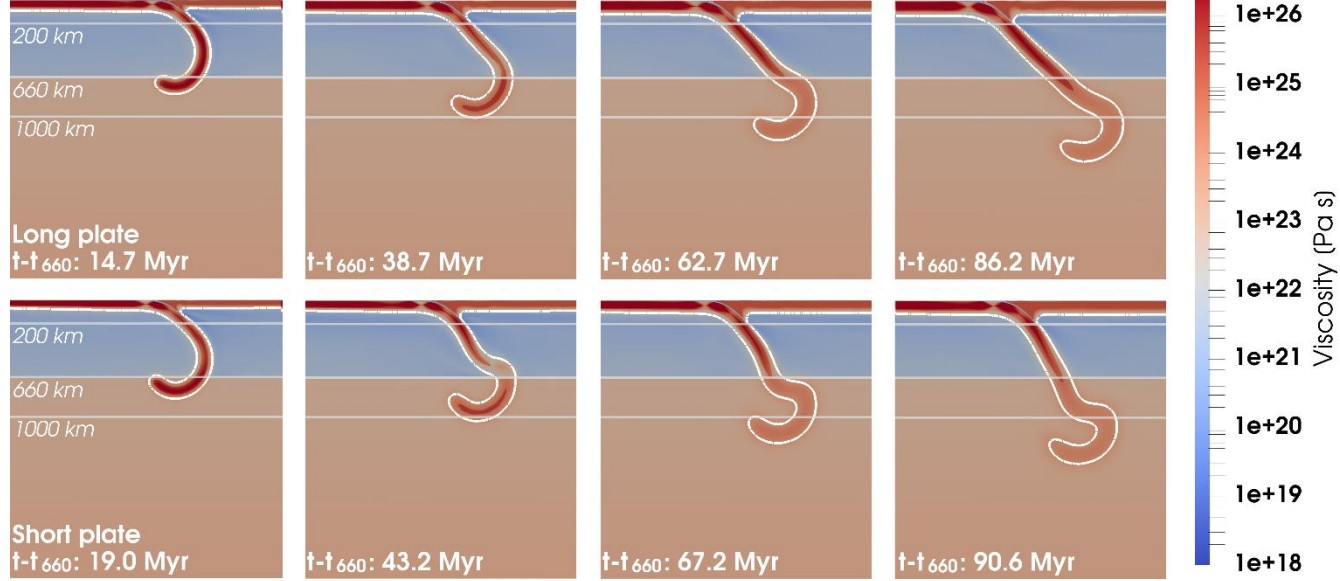


**Figure B1 (continue)**



## Combined weak asthenosphere and strong lithosphere (factor 2)

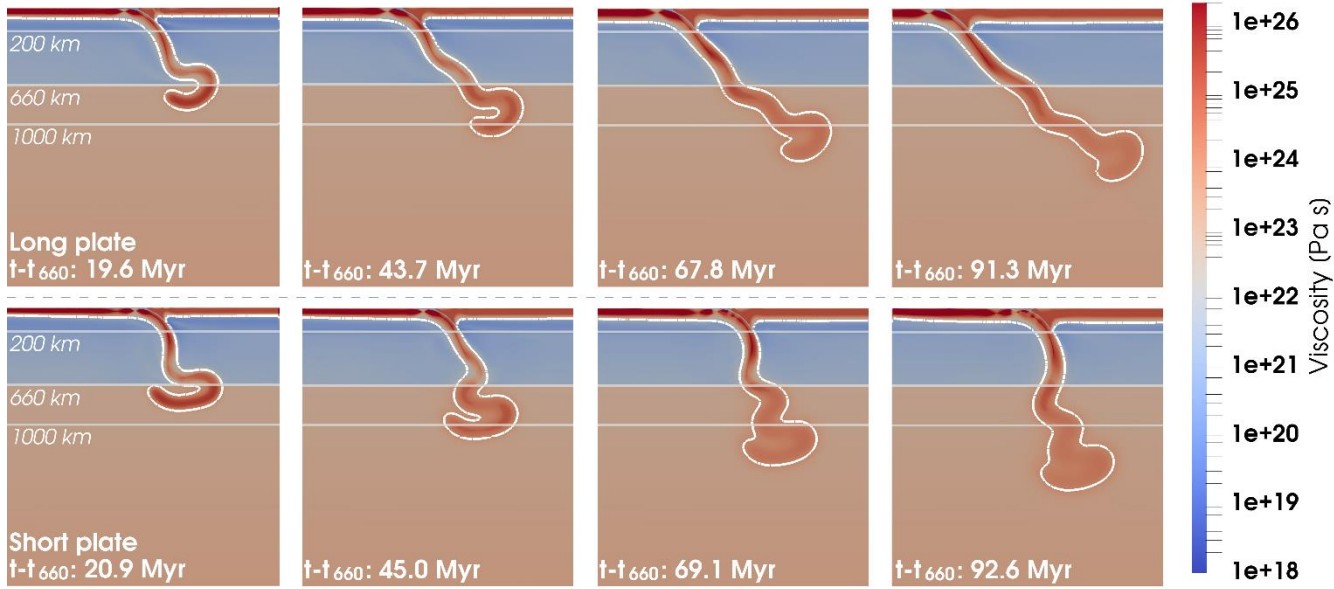

## Combined weak asthenosphere and strong lithosphere (factor 10)

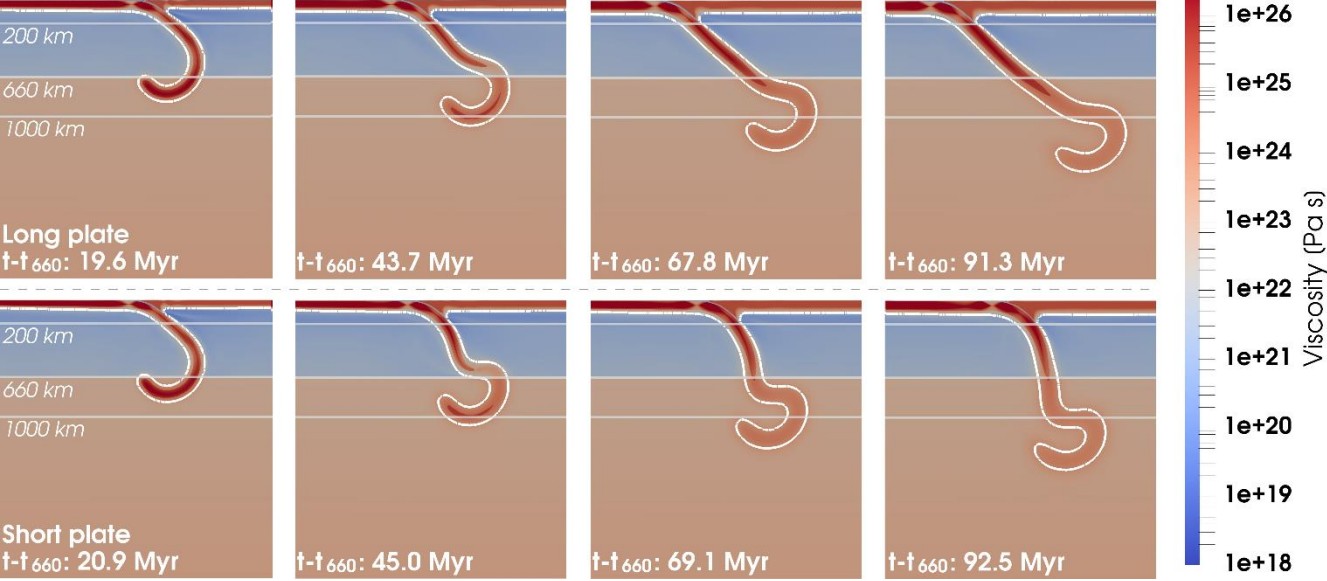

**Figure B1 (continue)**



# Self consistent rheology (factor 2)

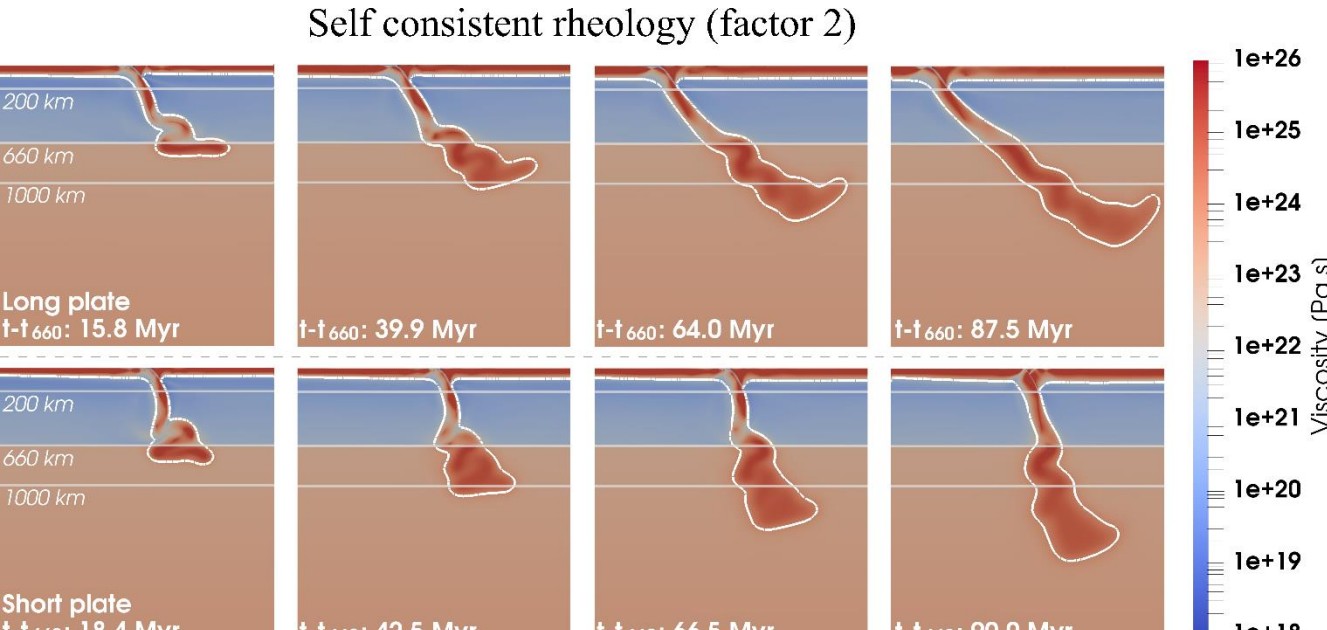

**Figure B1 (continue)**

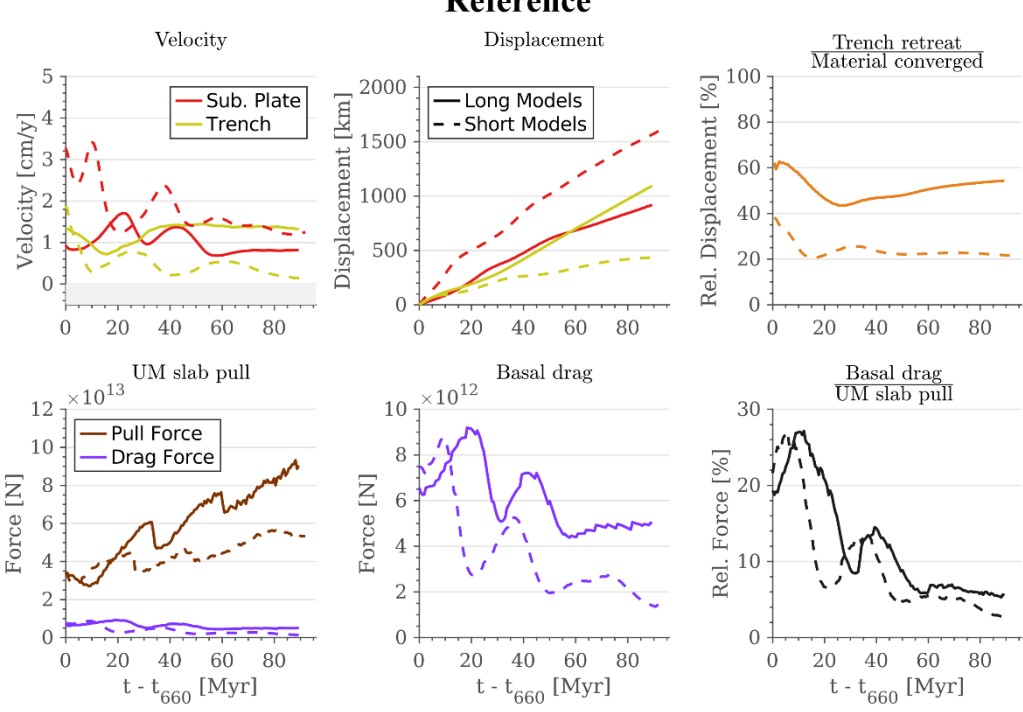

**Figure B2: Evolution graphs (as Fig. 4) for all models**





**Figure B2 (continue)**





# Strong lithosphere (factor 10)

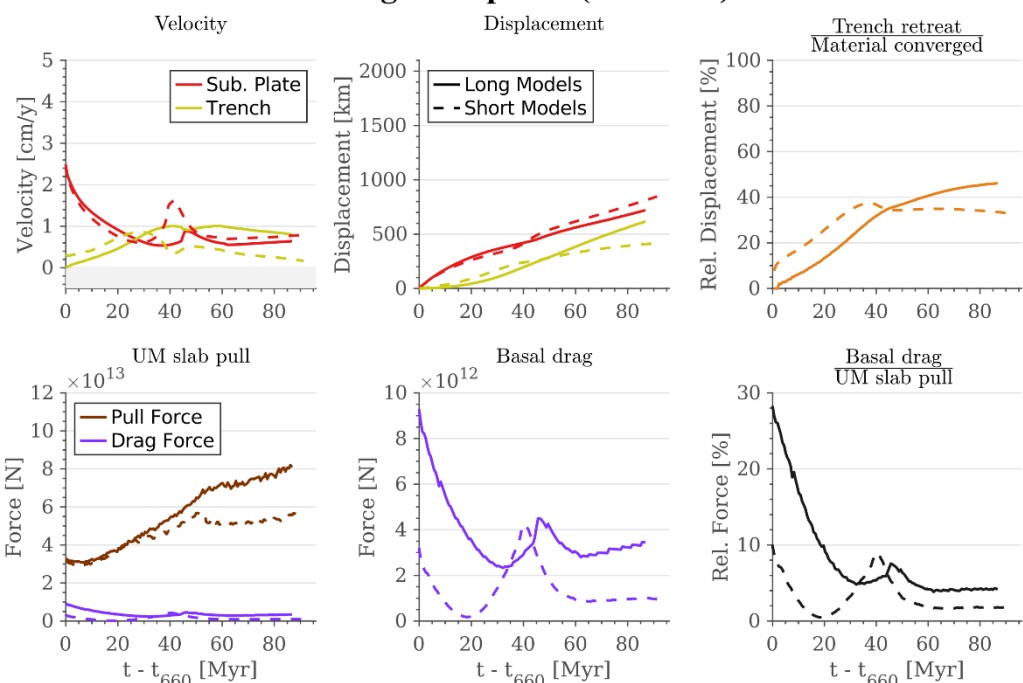

# Combined weak asthenosphere and strong lithosphere (factor 2)

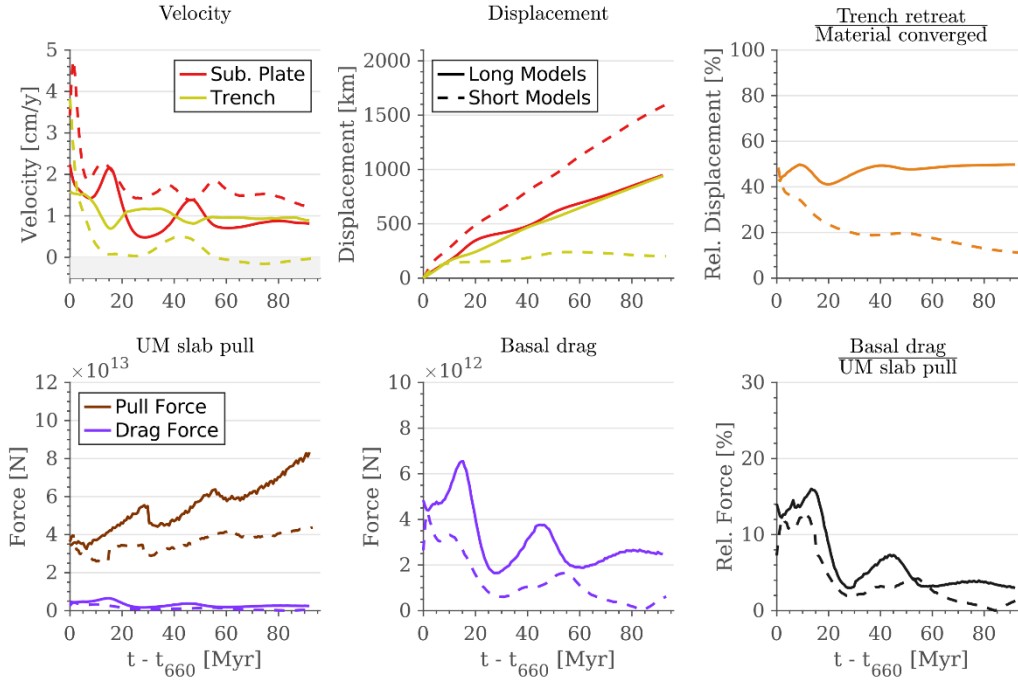

**Figure B2 (continue)**





## Combined weak asthenosphere and strong lithosphere (factor 10)

## Self consistent rheology

**Figure B2 (continue)**



## Appendix C, subduction zone parameters

We used GPlates (Müller et al., 2018) with the plate reconstruction of Müller et al. (2016) to make an updated compilation of the average velocities, average age at the trench and size of the subducting plates for major subduction zones through the Cenozoic (0-60 Ma). This was done for all trenches previously studied by Sdrolias and Müller (2006) who considered the Andean subduction zone, Central-North Farallon subduction systems, subduction below Alaska and the Aleutians, subduction below Japan-Kuriles-Kamchatka, subduction below Izu-Bonin-Marianas, below Tonga-Kermadec and below the

Sunda-Banda arc. To these we added the subduction of the Philippine Sea plate and the final stages of the subduction of the Izanagi and Kula plates. In processing this database, we considered all the Pacific subduction systems (i.e. Alaska-Aleutians, Japan-Kuriles-Kamchatka, Izu-Bonin-Marianas and Tonga-Kermadec) as a single Pacific system.

The relevant trenches were identified by extracting the global subducted segments of all plate polygons in GPlates. All trenches were sampled at 50 km intervals, and the coordinates were outputted and then plotted to select those belonging to

the major subducting plate systems listed above. For each selected system, age and velocities along the trench were plotted every 10 Myr to check that all data made sense, and to remove edge points where these were clearly anomalous from the rest of the trench (e.g. because of an anomalous age or convergence direction). We evaluated the point velocity of the subducting plate, of the overriding plate as well as convergence velocity and direction to select the segments to analyse. Maps showing the trench segments included in our analysis at each stage can be found in Fig. C1, where the sampling points along the

trenches are also coloured according to the subducting plate absolute velocity (in the moving hotspot reference frame of O'Neill et al.) and age. These maps illustrate the evolving set of subduction systems through the Cenozoic as well as the variability of age and velocity along each trench. Aside from some edge points, we excluded from our analysis the part of the South American trench at 40 and 50 Myr which was south of the Antarctic ridge and the Cocos subduction system at 20 Ma, because according to the plate motion model there is limited convergence along most of the trench at this time. Finally, we

considered for our analysis the mean value of the velocities and trench age for each subducting system, and the standard variation of the mean value as the uncertainty.

There is quite some variation of both velocity and age along each trench, and each trench has a unique tectonic evolution. Nonetheless, the lack of a trend between velocity and size and the overall correlation between size and age are general features of the Cenozoic set of subduction zones, not dependent on including or excluding one or the other system, or on

somewhat different definitions of each system or on with whether the total subducting plate velocity or only the normal component of velocity is used.





**Figure C1: Evolution of global trenches considered in the analysis of the Cenozoic subduction systems, in intervals of 10 Myr. 3 maps used for each time interval, showing the trenches used for the analysis (top), absolute subducting plate velocity in hotspot reference frame (O'Neill et al., 2005; middle) and age at the trench (bottom). Grey lines represent other boundaries of the subducting plates. The background light grey represents present day global coastlines. Data is based on Müller et al. (2016) and processed using GPlates and Cartopy (Met Office, 2015; Müller et al., 2018).**




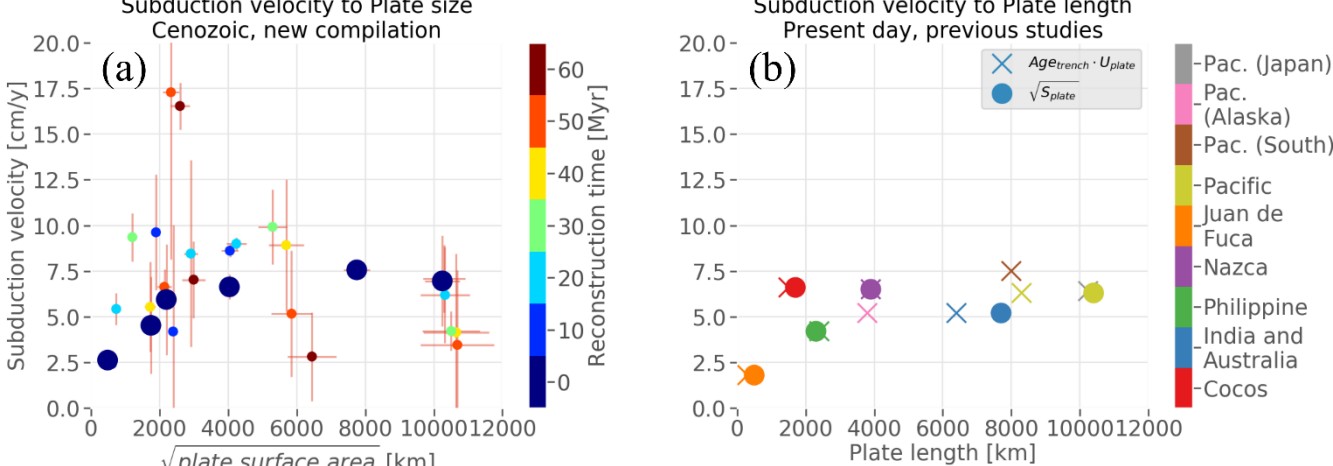

**Figure C2: Velocity of subducting plates to plate length throughout the Cenozoic (a) and at present day (b). Circles represent**
**length calculated as the square root of the surface area of the plate and X markers indicate length as aa multiplication of**
**subduction velocity and trench age. Bold markers indicate present day values. Cenozoic dataset is based on plate reconstruction**
**from Müller et al. (2016) and analysed using GPlates (Müller et al., 2018). Present day data from Conrad and Hager (1999),**
**Schellart et al. (2007) and Sdrolias and Müller (2006). Velocities are calculated in lower mantle (plume) reference frame (O'Neill et**
**al., 2005). Further details in Fig. 8 and in the main text.**

**Code availability**

Fluidity is publicly available online at https://fluidityproject.github.io/.

**Author contribution**

Lior Suchoy is the main researcher of the project.

Saskia Goes supervised the project and took part in every aspect of it.

Benjamin Maunder assisted in designing the models and provided creative solutions for technical difficulties, helped in

analysing the results of the models and provided the code used for the Self-Consistent Rheology model.

Fanny Garel provided the design of the model which was used as the basis of the Reference model and commented on the

interpretation and limitation of the results of the model.

Rhodri Davies designed the Stokes branch of Fluidity, assisted in the design and setup of the Reference model and

commented on the technical aspects of paper.





**Acknowledgments**

Lior Suchoy was funded by an EPSRC DTP studentship (EP/N509486/1), Ben Maunder and Saskia Goes were funded under
NERC grant NE/K010743/1. DRD acknowledges support from the Australian Research Council, under grant number
DP170100058.

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
