# Peer review of "Effects of basal drag on subduction dynamics from 2D numerical models"

_Solid Earth, 2020_

## Referee Comment (RC1) · Anonymous Referee #1 · 5 Aug 2020

General comments Suchoy and co-authors use numerical models to study the effect of basal drag on subduction dynamics and uses them to explain the lack of plate velocity-size correlation. The manuscript is well written and the models are well constructed. Overall, I think this is an interesting study that deserves publication after minor revisions.

Specific comments Many times in the introduction the authors talk about the viscosity of the asthenosphere and the ratio between lithosphere-asthenosphere viscosity as 'high' or 'low' or 'too low'. Since this is a crucial part of the paper, I think the author should describe this in a more quantitative manner (i.e., mention some estimates with actual numbers suggested by the cited literature). This is true also throughout the rest of the manuscript, even in the results where the viscosity values are easily to extract from the

models (e.g., lines 223-224). The same for the conclusions: e.g. line 441-443 "Models with a low-viscosity asthenosphere do reduce the contribution of trench motion to plate convergence to more Earth-like values, as observed in previous studies". How low? Which values range are we talking about?

Following the previous point, I would be curious to see the evolution of the viscosity of the asthenosphere in Fig. 4 (or in the Supplementary). The reason for this is that I have a few doubts on (1) the effect of the used methodology and (2) the effect of plate velocity on the viscosity of the asthenosphere and a figure that shows it would likely clear them out. In particular: (1) Regarding the methodology. The viscosity of the asthenosphere is weakened by multiplying the computed viscosity by a factor of 0.5. However, the computed viscosity is a combination of diffusion, dislocation and Pierels creep, thus, it is strain dependent. When lowering the viscosity by a factor 0.5, the stresses in the asthenosphere will be lower too and the viscosity will want to be higher again at the next time step. Is that what happen? Or does it more or less stabilize? (2) Several times the authors say that because the velocity of a plate increases then the basal drag increases too. However, wouldn't the viscosity of the asthenosphere decrease with high plate velocities due to larger deformation? If so, the basal drag would decrease too or is there something else in Eq. 1 that also changes to compensate a decrease of EtaAst?

I do not fully understand the reasoning in the Discussion about the correlation between plate size and age at the trench (Fig. 8B). A larger plate would have ridges further away from trenches (e.g., Pacific vs. Cocos) and therefore more likely to have older lithosphere at trenches, could it not be 'that simple' without the need to use the basal drag as explanation?

The authors should discuss how their results and conclusions could be affected by viscous anisotropy? From Becker and Kawakatsu, GRL, 2011: "One of the major limitations of our study is that we only considered a few instantaneous flow examples for which the influence of anisotropy may overall be negligible. This only indirectly

addresses more complex, evolving scenarios such as changes in plate motions, or plate boundary dynamics, where mechanical anisotropy may well be relevant." Viscous anisotropy has the potential to have an important effect when looking at plate velocities during the Cenozoic as the authors do here because changes in plate motion direction would change the orientation of the anisotropy and could contribute to change the asthenosphere viscosity (hence, basal drag). I understand that this cannot be included in the calculations, but I think it deserves to be mentioned and discussed.

I think the first paragraph of Conclusions belongs to the Discussion. Consider moving it there.

Eq. 1 and 2. How is the lithosphere defined? Is it defined by the 1100 degC isotherm? Please specify it, since it matters for parameters like hLit, hAst, and Sslab. (I found the answer later on in the lines 144-145, but the authors might want to either repeat it or move it here where the variables of the equations are explained). At what depth is Sslab taken? And is it an horizontal section or perpendicular to the slab? Given the importance of these calculations for the study, I would suggest to have a figure with a schematic cartoon of a model showing where and how all the variables used in Eq. 1 and 2 are taken. It could go in the main manuscript, in the supplementary material, or merged with Fig. 2.

Table 1. Why is the slab pull force (and the basal drag) for the Cocos plate one order of magnitude larger than for the Pacific? Shouldn't it be the opposite? Please check your calculations. U is the plate velocity, what is deltaU (used in Eq.1)? S should be Splate (as referred to in the text). Is Vslab = Sslab*Ltrench or Sslab*Ltrench*Wslab?

Fig. C1: at the moment this figure is confusing. A better layout could be with each column showing a parameter. And also using more distinct colormaps for age and subduction velocity (now they are very similar). Also the colours of the different subduction zones are different from those used in the other figures (Fig. 1, 8, C2). When possible, I would suggest to keep them the same.

[Figure]

Technical corrections Line 331: odd sentence, rephrase. Line 338: Other approximations in addition to what? Line 401: "tend to be mostly fall around..." delete 'be' or 'fall'

---

## Referee Comment (RC2) · 14 Aug 2020

*Review of*
**"Effects of basal drag on subduction dynamics from 2D numerical models"**
**by Suchoy et al.**

The authors have performed 2D numerical experiments of plate subduction in order to assess the role of basal drag on Earth's tectonic plate velocity-size correlation (if there is any), subduction behaviour (e.g. trench advance/retreat) and slab morphology. The numerical models are self-consistent, well tested for potential artefacts (e.g. box size) and well parameterized (e.g. rheological parameters). The text is well written. The research question addressed in the paper, which is understanding how mobile the plates are above the viscous asthenosphere during plate subduction and how this impacts the subduction system in general, is very appealing to geodynamics community, and the findings certainly have the potential to well contribute to geosciences in general only if the analysis of the numerical models is made more carefully and the arguments in the manuscript are widely revised based on the new analysis. Overall, the authors attempt to convey a number of interesting ideas in tandem, however, in its present form, the manuscript lacks a certain level of clarity and focus. To be more explicit, I provide my comments as below.

Ömer F. Bodur

I do find the arguments about linking slab strength to trench advance/retreat convincing and important. However, I do have concern that the analysis of the numerical models might have been done with partial ignorance of the forces (mostly resistive) acting on the subducting plate (Solomon and Sleep, 1974; Forsyth and Uyeda, 1975), henceforth limiting our understanding of how basal drag *solely* affects the whole plate subduction system. For example, the drag around the sinking slab (i.e. slab resistance) must also be acting along with basal drag and this has not been considered in the analysis so that the

readers cannot separate the contribution of each. Therefore, it's not clear if it's the basal drag component itself driving the changes in plate velocity and slab morphology.

Similarly, once the slab dip is small and there is longer slab sinking into the mantle, analysis of *relative* significance of basal drag to the slab pull can be misleading because slab resistance is not taken into account with the increased slab area (i.e. slab interface in 2D) although it was considered for slab pull. The arguments of force balance cannot be reduced only to basal drag and slab pull as other forces also act and vary in time in the numerical model.

Equally important, some of the model results contradict with what the paper proposes throughout the text. For example, in the numerical experiments, a decrease in asthenosphere viscosity results in an increase in relative basal drag right after (for about 10 Myr) the plate has sunk to 660 km depth (comparing light red line with green line in Fig 6d). This is contrary to what is expected and needs clarification well before addressing other effects of basal drag (e.g. slab morphology).

The authors use a derived equation to estimate the slab pull and basal drag forces over time. These equations are (Eqn. 1, 2), most likely, valid only for iso-viscous plate and asthenosphere, and therefore may not be suitable to apply on the numerical modelling results. The authors also need to be cautious about necking of the subducting plate which results in significantly lower viscosities at the subducting plate as can be seen in Figure 3. This means, the slab pull force cannot act efficiently on the unsubducted part of the plate, hence an interpretation of the relative strength of basal drag and slab pull may become misinforming.

**Abstract:** It's worth briefly indicating why the force balance in subduction dynamics is incompletely understood (e.g. fails to explain plate velocity?) I think the force balance, or the method itself, is not to blame, but the contributions to the force balance by different forces are quite uncertain. This should be made clear.

**[Lines: 140-150]:** The lithosphere has a number of definitions and one of which is by viscosity profile (e.g. (Conrad and Molnar, 1997). In numerical models, the viscosities will vary and the effective lithospheric thickness that you have used in the estimation of basal drag will also change accordingly (Bodur and Rey, 2019).

**[Lines 158-160]:** Please justify why avoiding any slab detachment during subduction is favoured. If the model results are only applicable to plates not showing slab detachment, then this should be mentioned early in the text or in the abstract. It's important to acknowledge that slab detachment has been used to explain important features of the Earth (Göğüş and Psyklywec, 2008; Duretz and Gerya, 2013; Hacker and Gerya, 2013)

**[Eqn1]:** Please provide the derivation of the equation for basal drag and/or the page # of the citation you provided.

**Section 4.3 [Lines 364-365]:** The slab dip you consider here is quite higher than numerical models show. Why? It's also unclear what sort of data you have used to calculate the slab pull and basal drag estimates in Table 1. Please be more specific so that one can derive the same results individually for further reference. Also, for different plates, the asthenosphere viscosity ( $\eta_{asth}=10^{19}$ Pa·s) is not necessarily the same, so you may need to consider different viscosities, at least mentioning about it.

**[Line 391]:** Although they can be correlated, Fig. 8b doesn't show subduction zone length, but the plate size vs. plate age at trench. It's better to be more specific.

**[Lines: 399-401]:** The correlation is weak already (based on error bars and scattered points in Fig. 8a), and the argument on explaining an already weak correlation "*at least in part*" is making this sentence more confusing for readers. I recommend restructuring those lines.

*Additional Comments:*

**[Line 156]:** 1024 Pa·s – 1025 Pa·s should be changed to $10^{24}$ Pa·s – $10^{25}$ Pa·s.

**[Line 383]:** Fig. B1 needs to be changed to Fig. C1.

*References:*

Bodur, Ö. F. and Rey, P. F.: The impact of rheological uncertainty on dynamic topography predictions, Solid Earth, 10(6), 2167–2178, doi:10.5194/se-10-2167-2019, 2019.

Conrad, C. P. and Molnar, P.: The growth of Rayleigh-Taylor-type instabilities in the lithosphere for various rheological and density structures, , 95–112, 1997.

Duretz, T. and Gerya, T. V.: Slab detachment during continental collision: Influence of crustal rheology and interaction with lithospheric delamination, Tectonophysics, 602, 124–140, doi:10.1016/j.tecto.2012.12.024, 2013.

Forsyth, D. and Uyeda, S.: On the Relative Importance of the Driving Forces of Plate Motion, Geophys. J. R. Astron. Soc., 43, 163–200, 1975.

Göğüş, O. H. and Psyklywec, R. N.: Mantle lithosphere delamination driving plateau uplift and synconvergent extension in eastern Anatolia, Geology, 36(9), 723–726, doi:10.1130/G24982A.1, 2008.

Hacker, B. R. and Gerya, T. V: Paradigms, new and old, for ultrahigh-pressure tectonism, Tectonophysics, 603(0), 79–88, doi:http://dx.doi.org/10.1016/j.tecto.2013.05.026, 2013.

Solomon, S. C. and Sleep, N. H.: Some simple physical models for absolute plate motions, J. Geophys. Res., 79(17), 2557, doi:10.1029/JB079i017p02557, 1974.

---

## Author Response (AR1)

Dear Editor,

Thank you for the chance to revise our paper.
These are the main changes we made, all in response to questions from both reviewers. Detailed responses to each of the reviewers' comments can be found below.

- To illustrate the evolution of asthenosphere viscosity and velocity to help the discussion of how drag evolves as these two parameters evolve, we added Figure S2-3 (Note that we moved all the appendices to the supplementary material).

- We added further motivation for the used equation for estimating drag, for evaluating drag relative to the main driving force (slab pull), and more information on how parameters for drag estimates in the model were measured (including information marked on the new Figure S2-3).

- We explain better that the correlation between plate size and age is a feature of the Cenozoic plate configuration, not a causal relationship.

- We fixed some copying errors in the table entries of the analytical estimates of Pacific and Cocos plate drag.

We thank **Referee #1** for their positive feedback and their constructive comments. We considered each comment and revised the manuscript accordingly. Please see below our responses to each comment (in blue, with changes made in *italics*) and the annotated manuscript for the revisions.

1. *Many times in the introduction the authors talk about the viscosity of the asthenosphere and the ratio between lithosphere-asthenosphere viscosity as 'high' or 'low' or 'too low'. Since this is a crucial part of the paper, I think the author should describe this in a more quantitative manner (i.e., mention some estimates with actual numbers suggested by the cited literature). This is true also throughout the rest of the manuscript, even in the results where the viscosity values are easily to extract from the models (e.g., lines 223-224). The same for the conclusions: e.g. line 441-443 "Models with a low-viscosity asthenosphere do reduce the contribution of trench motion to plate convergence to more Earth-like values, as observed in previous studies". How low? Which values range are we talking about?*
   We agree that it is useful to add some more numbers. Many previous models considered ratios of effective lithosphere-asthenosphere viscosities of a factor 100-1000 (with reference asthenospheric viscosities of $10^{21}$ to $10^{20}$ Pa s). Weak asthenosphere models tend to have viscosity ratios of $10^3$ to $10^4$ (for reference asthenospheric viscosities of $10^{20}$ to $10^{19}$ Pa s), which when these are the outcome of non-linear composite rheology still allows for viable subduction. *We now added these numbers plus references to the introduction for context (line 53-55, 71-73 and 85). The values used in the models are discussed and motivated with references in the last paragraph of the 'Methods' section (lines 157-166). We added a new supplementary figure (Fig. S2.3) that displays the evolution of viscosity through time for the long and short reference models along the profile that is used for basal drag calculations. And we include numbers for the viscosity ratios in our models in the conclusions (lines 462 and 468)*

2. *Following the previous point, I would be curious to see the evolution of the viscosity of the asthenosphere in Fig. 4 (or in the Supplementary). The reason for this is that I have a few doubts on (1) the effect of the used methodology and (2) the effect of plate velocity on the viscosity of the asthenosphere and a figure that shows it would likely clear them out. In particular: (1) Regarding the methodology. The viscosity of the asthenosphere is weakened by multiplying the computed viscosity by a factor of 0.5. However, the computed viscosity is a combination of diffusion, dislocation and Pierels creep, thus, it is strain dependent. When lowering the viscosity by a factor 0.5, the stresses in the asthenosphere will be lower too and the viscosity will want to be higher again at the next time step. Is that what happen? Or does it more or less stabilize? (2) Several times the authors say that because the velocity of a plate increases then the basal drag increases too. However, wouldn't the viscosity of the asthenosphere decrease with high plate velocities due to larger deformation? If so, the basal drag would decrease too or is there something else in Eq. 1 that also changes to compensate a decrease of EtaAst?*
   (1) Indeed, the viscosity is strain-rate dependent. The lowering of viscosity by a factor 0.5 leads to increased velocities. Increased velocities are associated with increased strain rate, and lead to a feedback that results in further lowering (rather than increasing) of viscosity. The net effect of the scaling is a reduction of viscosity by more than the applied scaling factor. *We write: "The compound effect of applying the 0.5 reduction factor and consequent strain-rate weakening is an order of magnitude decrease in asthenospheric viscosity compared to the reference models."*
   (2) The viscosity of the asthenosphere will tend to decrease with higher velocity (if this leads to higher strain rates), reducing the basal drag. However, the direct effect of increased velocity is an increase in drag, which can dominate over the effect of the lowering of viscosity. This happens in the reference case short-plate models, around the time the plates reach 660 (t660), and in the weak asthenosphere long-plate models around t660. The two effect of increased velocity, direct and on viscosity, is also the reason that the plate velocity graph and the basal drag graph do not always show the same trends. *We now discuss this non-linearity in our presentation of the reference models and added a graph showing the evolution of the viscosity and velocity to the supplementary material: "Note that due to the*

*nonlinear feedbacks in the models, there are about 10 Myr right after the plates reach the ULMB where the short plate experiences a higher drag than the long plate. These relatively high drag values for the short plate are a consequence of the high velocities that the short plate attains in response to the initially significantly lower drag (Fig. S2.3)."*

3. *I do not fully understand the reasoning in the Discussion about the correlation between plate size and age at the trench (Fig. 8B). A larger plate would have ridges further away from trenches (e.g., Pacific vs. Cocos) and therefore more likely to have older lithosphere at trenches, could it not be 'that simple' without the need to use the basal drag as explanation?*

The correlation of size and age is an observation. Such a correlation might be expected if spreading rate is relatively constant. However, closer examination of the plate reconstruction reveals a more complex picture of changing spreading rates, subducting ridges, changing trench retreat rates and plate geometry that eventually determine plate size and trench age.

We use this observation to argue that larger plates tend to have a stronger driving force. Increased basal drag with increasing plate size can explain that larger plates are not observed to reach systematically higher velocities. This explanation only applies to the Cenozoic where plate size and age of the subducting plate are correlated.

*We changed some of the phrasing in the revised manuscript to make it clear these are separate observations. In the discussion we write: "A consequence of the observed correlation of average age at the trench and plate size is that plates with a stronger slab pull tend to also have larger surface area and hence a stronger resisting basal drag. This provides a mechanism that offsets the velocity-enhancing effect of larger driving forces of old plates, and could explain the observation that velocities of subducting plates on Earth, today and throughout the Cenozoic, do not increase with age but tend to be mostly stable around 8-10 cm/yr".*

*We also write in the discussion: "Note that the relation between age and size is not a causal relationship, but a feature of the plate configuration that has dominated most of the Cenozoic. Early in the Cenozoic, there are several cases that deviate from the buffered velocity trend. At the start of the Cenozoic, much of the subduction surrounding the Pacific plate consumed relatively young lithosphere, even though the Pacific plate itself was already large in size (horizontally aligned points at the top of the area–age trend in Fig. 8b). Other early Cenozoic deviations include very high velocities of the last remnants of the Izanagi and Kula plates (points with area of about $2500^2$ $km^2$ and velocities around 17 cm/yr in Fig. 8a) and low velocities of Farallon plate (area about $6000^2$ $km^2$ in Fig. 8a)".*

*In the conclusions, we reworded the last paragraph to: "Based upon an analysis of a Cenozoic plate motion reconstruction (Müller et al.,, 2016), we suggest that the reason that most plates move at velocities around 8-10 cm/yr is because the plate configuration during this era was such that plate size correlated with plate age at the trench, i.e., both driving and resisting forces increased together. Note however, that this correlation between size and age is not causal and may not have existed in other times of Earth history. As a result, during the Cenozoic, the increase in basal drag more or less balanced the increase in plate velocity induced by increased slab pull with increasing age. Such co-variations between plate velocity, age and size should be considered in regional models of subduction systems"*

4. *The authors should discuss how their results and conclusions could be affected by viscous anisotropy? From Becker and Kawakatsu, GRL, 2011: "One of the major limitations of our study is that we only considered a few instantaneous flow examples for which the influence of anisotropy may overall be negligible. This only indirectly addresses more complex, evolving scenarios such as changes in plate motions, or plate boundary dynamics, where mechanical anisotropy may well be relevant." Viscous anisotropy has the potential to have an important effect when looking at plate velocities during the Cenozoic as the authors do here because changes in plate motion direction would change the orientation of the anisotropy and could contribute to change the asthenosphere viscosity (hence, basal drag). I understand that this cannot be included in the calculations, but I think it deserves to be mentioned and discussed.*

*We now clarify that other factors may contribute to viscosity when we introduce our choice of rheology in the methods: "We use a temperature-, pressure- and strain-rate dependent composite rheology, which naturally results in a lithosphere and asthenosphere. Other factors may contribute to the contrast between lithosphere and asthenosphere viscosity, e.g. hydration, partial melt and anisotropy (e.g., Hirth and Kohlstedt, 1996; Becker and Kawakatsu, 2011). Here, we consider diffusion and dislocation creep mechanisms, combined with a low-pressure yield-stress mechanism to approximate brittle failure and an approximation of Peierls low-temperature plasticity at high pressure (e.g. Čížková et al., 2002; Garel et al., 2014). Equations and parameters for the reference cases are as in Garel et al. 2014 and are given in Table S1.1. We vary parameters, as discussed below, to evaluate different relative lithosphere and asthenosphere viscosities."*

5.  *I think the first paragraph of Conclusions belongs to the Discussion. Consider moving it there.*
    The first paragraph of the conclusions is a recap of the motivation for the study previously outlined in the introduction. We think it belongs in the conclusions section rather than the discussion section but hope that with our other clarifications of the introduction, discussion and conclusions this organisation makes more sense.

6.  *Eq. 1 and 2. How is the lithosphere defined? Is it defined by the 1100 degC isotherm? Please specify it, since it matters for parameters like hLit, hAst, and Sslab. (I found the answer later on in the lines 144-145, but the authors might want to either repeat it or move it here where the variables of the equations are explained). At what depth is Sslab taken? And is it an horizontal section or perpendicular to the slab? Given the importance of these calculations for the study, I would suggest to have a figure with a schematic cartoon of a model showing where and how all the variables used in Eq. 1 and 2 are taken. It could go in the main manuscript, in the supplementary material, or merged with Fig. 2.*
    The lithosphere is defined as the part of the model that is colder than 1100°C. This does mean the thickness of the lithosphere and asthenosphere evolves with time. It is mentioned in the 'Model set-up' section, on Figure 2 and in the caption for Figure 2. $S_{slab}$ is calculated as the surface of a rectangle with one side the length of the slab (*as explained in lines 204-206*), and the other side the width of the slab, taken as a horizontal line across the slab at 220 km depth. *We now improved the description of the parameters of Eq. 1 and Eq. 2 in the revised manuscript and added a figure in the supplementary material (Figure S2.3) where we also mark where some of the key parameters were measured.*

7.  *Table 1. Why is the slab pull force (and the basal drag) for the Cocos plate one order of magnitude larger than for the Pacific? Shouldn't it be the opposite? Please check your calculations. U is the plate velocity, what is deltaU (used in Eq.1)? S should be Splate (as referred to in the text). Is Vslab = Sslab\*Ltrench or Sslab\*Ltrench\*Wslab?*
    Thanks for catching this error. There was a copying error in the table (the values of the forces were copied to the wrong row in the table). Vslab is calculated as Lslab\*Ltrench\*Wslab where Lslab is assumed to be 700 km (lines 364-365). After re-examining the calculation, we found that there was a missing factor 2 for the drag force. *We corrected the mistake, the calculation, the corresponding text and expanded the description of Vslab in the revised manuscript.*

8.  *Fig. C1: at the moment this figure is confusing. A better layout could be with each column showing a parameter. And also using more distinct colormaps for age and subduction velocity (now they are very similar). Also the colours of the different subduction zones are different from those used in the other figures (Fig. 1, 8, C2). When possible, I would suggest to keep them the same.*
    *We changed the figure in the revised manuscript according to the comment.*

9.  *Technical corrections Line 331: odd sentence, rephrase. Line 338: Other approximations in addition to what? Line 401: "tend to be mostly fall around…" delete 'be' or 'fall'*
    *We changed the sentences in the revised manuscript according to the comments.*

We thank Ömer Bodur (**Referee #2**) for the detailed feedback and comments. We considered the comments carefully and revised the manuscript accordingly. Please find our replies to the comments in the following section (in blue, with changes made in *italics*) and see the annotated manuscript for changes to the text.

1.  *I do find the arguments about linking slab strength to trench advance/retreat convincing and important. However, I do have concern that the analysis of the numerical models might have been done with partial ignorance of the forces (mostly resistive) acting on the subducting plate (Solomon and Sleep, 1974; Forsyth and Uyeda, 1975), henceforth limiting our understanding of how basal drag solely affects the whole plate subduction system. For example, the drag around the sinking slab (i.e. slab resistance) must also be acting along with basal drag and this has not been considered in the analysis so that the readers cannot separate the contribution of each. Therefore, it's not clear if it's the basal drag component itself driving the changes in plate velocity and slab morphology.*
    *Similarly, once the slab dip is small and there is longer slab sinking into the mantle, analysis of relative significance of basal drag to the slab pull can be misleading because slab resistance is not taken into account with the increased slab area (i.e. slab interface in 2D) although it was considered for slab pull. The arguments of force balance cannot be reduced only to basal drag and slab pull as other forces also act and vary in time in the numerical model.*

    We agree that the force balance cannot be reduced to basal drag and slab pull. However, we make a different argument in the text. We show, throughout the evolution of the model, how much of the slab pull is resisted by basal drag. We do not make the point that the evolution of velocity and slab morphology is controlled solely by these two forces, nor do we try to show the full force balance acting on the plate and slab throughout the evolution. By referring to the ratio of basal drag to slab pull, we can compare the significance of basal drag in the subduction process to other forces, such as viscous dissipation and ridge push, which have been examined against slab pull in previous studies. *We discuss the range of forces that contribute in the second paragraph of the introduction and have now worded this more clearly when we introduce the equations used to estimate basal drag and slab pull: "To evaluate the relative importance of basal drag in the overall force balance, we estimate basal drag below the subducting plate for each time step in the models, and compare it to the main driving force, slab pull."*

2.  *Equally important, some of the model results contradict with what the paper proposes throughout the text. For example, in the numerical experiments, a decrease in asthenosphere viscosity results in an increase in relative basal drag right after (for about 10 Myr) the plate has sunk to 660 km depth (comparing light red line with green line in Fig 6d). This is contrary to what is expected and needs clarification well before addressing other effects of basal drag (e.g. slab morphology).*

    Indeed, the non-linear feedbacks in the models sometimes cause unexpected results. In this case, the lower viscosity of the asthenosphere leads to high velocities which by the time the plate reaches 660, mean that the long-plate model with lower viscosity asthenosphere, briefly (~ 10 Ma) experiences a higher drag than the reference case with higher asthenospheric viscosity at this same stage of the model. However, in spite of the complexities in how basal drag evolves, all results are consistent with the conclusion that basal drag is important.

    *We added a short discussion of this to the description of the Reference models in the Results section in the revised manuscript: "Note that due to the nonlinear feedbacks in the models, there are about 10 Myr right after the plates reach the ULMB where the short plate experiences a higher drag than the long plate. These relatively high drag values for the short plate are a consequence of the high velocities that the short plate attains in response to the initially significantly lower drag (Fig. S2-3)."*

3. *The authors use a derived equation to estimate the slab pull and basal drag forces over time. These equations are (Eqn. 1, 2), most likely, valid only for iso-viscous plate and asthenosphere, and therefore may not be suitable to apply on the numerical modelling results. The authors also need to be cautious about necking of the subducting plate which results in significantly lower viscosities at the subducting plate as can be seen in Figure 3. This means, the slab pull force cannot act efficiently on the unsubducted part of the plate, hence an interpretation of the relative strength of basal drag and slab pull may become misinforming.*

Indeed Eq. 1 and 2 approximate the forces. In theory one could integrate over velocity, viscosity and density fields for more detailed measurements. However, this too would involve assumptions in choosing the boundary of the driving lithosphere and resisting asthenosphere, and a more non-linear dependence on those choices. We found the approximate solution illustrates model behaviour well enough. Due to our adaptive mesh, we are able to resolve the strong core of the plate even if it is thin. Note that the minimum element size is 0.4 km, as noted in the text, which allows us to resolve a thin strong core, which may not be clearly reflected on the colour scale in Fig. 3 (but illustrated clearly in figures in Garel et al., 2014). This means that stress is always quite effectively transmitted from the surface plate to the slab and from the slab to the plate. In all of the results we analysed, we found that the viscosity in the bulk of the slab was at least an order of magnitude higher than the surrounding mantle

*We added a mention of this where the equations are introduced in the revised manuscript. "Note that the expression for $F_{SP}$ assumes stresses are effectively transmitted from the slab to the plate, which is appropriate because the adaptive grids of our models ensure we always resolve the strong slab core, even when it is quite thin (Garel et al., 2014)".*

4. *Abstract: It's worth briefly indicating why the force balance in subduction dynamics is incompletely understood (e.g. fails to explain plate velocity?) I think the force balance, or the method itself, is not to blame, but the contributions to the force balance by different forces are quite uncertain. This should be made clear.*

*We reworded this sentence to "Subducting slabs are an important driver of plate motions, yet the relative importance of different forces in governing subduction motions and styles remains incompletely understood."*

5. *[Lines: 140-150]: The lithosphere has a number of definitions and one of which is by viscosity profile (e.g. (Conrad and Molnar, 1997). In numerical models, the viscosities will vary and the effective lithospheric thickness that you have used in the estimation of basal drag will also change accordingly (Bodur and Rey, 2019).*

That is indeed the case in our models. We use the 1100°C isotherm as the base of the lithosphere. The depth and shape of the isotherm changes with time and the thickness of the lithosphere changes with it. To account for that in the drag calculation, we measure the thickness at a given from the trench at each time step. *The process is detailed in lines 200-204, and new supplementary Figure S2.3 illustrates the evolution of various parameters, including the thickness of the lithosphere, through time, for the reference model.*

6. *[Lines 158-160]: Please justify why avoiding any slab detachment during subduction is favoured. If the model results are only applicable to plates not showing slab detachment, then this should be mentioned early in the text or in the abstract. It's important to acknowledge that slab detachment has been used to explain important features of the Earth (Göǧüş and Psyklywec, 2008; Duretz and Gerya, 2013; Hacker and Gerya, 2013)*

Our aim in this paper is to model basic continuous subduction behaviour. Slab detachment is the terminal stage of subduction, which has been observed in different settings than ocean-ocean subduction (e.g. continental collision, subduction of continental fragments or seamounts, etc.

See examples in Hacker and Gerya, 2013). We therefore aim for a rheology that allows continued subduction rather than slab detachment, or subduction freeze.

*We added this explanation to the Model Set-Up section in the revised manuscript:* "With this range of lithospheric and asthenospheric viscosities, we can generate continuous Earth-like oceanic style subduction while avoiding immediate slab detachments which result from to lithospheric weakening due to high strain rates, or the stalling of subduction due to unattainable forces required for the bending of very strong lithosphere".

7. *[Eqn1]: Please provide the derivation of the equation for basal drag and/or the page # of the citation you provided.*

The equation is based on Eq. (6.24) in section 6.3, Page 419, Turcotte, D. L. and Schubert, G.: Geodynamics, 2nd ed., Cambridge University Press, Cambridge. Available from: http://dx.doi.org/10.1017/CBO9780511807442, 2002. *We added the section number to the reference in the text and include the doi link in the reference list.*

8. *Section 4.3 [Lines 364-365]: The slab dip you consider here is quite higher than numerical models show. Why? It's also unclear what sort of data you have used to calculate the slab pull and basal drag estimates in Table 1. Please be more specific so that one can derive the same results individually for further reference. Also, for different plates, the asthenosphere viscosity ($\eta_{asth}=10^{19}$ Pa×s) is not necessarily the same, so you may need to consider different viscosities, at least mentioning about it.*

This is partly the result of copying mistake. The dip used for the calculation was 70° rather than 80°. This dip angle was chosen as it is representative for dip angles observed on Earth (Lallemand et al. 2005). It should be noted that:

- Considering a single value of slab dip angle, as well as single value for the slab width and other parameters, for a plate as varied as the Pacific cannot provide more than a first order estimation.
- The difference between considering an angle of 70° and 50° is a factor of 1.2. This is a minor uncertainty considering the estimation of slab width and other values.
- As first order estimation of forces, we think it is justified to use a single value of asthenospheric viscosity.

*We corrected the error and added a statement that the calculation is used for first order approximation in the revised manuscript. And we explain that the dip used is a representative dip from a global compilation of subduction parameters.*

9. *[Line 391]: Although they can be correlated, Fig. 8b doesn't show subduction zone length, but the plate size vs. plate age at trench. It's better to be more specific.*

Fig. 8b presents the square root of the surface area of the plate vs. trench age. The square root has units of length. In Fig. 1, plate size is displayed in terms of a typical length calculated in two different ways. The way typical length is calculated is explained in the captions. *We changed the labels of Fig. 1 and 8b to "Plate size [typical length in km]". To avoid any confusion, we removed references to length when discussing 3D plates. It is only used when describing the results of the 2D models.*

10. *[Lines: 399-401]: The correlation is weak already (based on error bars and scattered points in Fig. 8a), and the argument on explaining an already weak correlation "at least in part" is making this sentence more confusing for readers. I recommend restructuring those lines.*

*We rephrased this part in the revised manuscript to make our intention clearer.* We explain that old plates tend to be large, and as a result their movement is resisted more. This provides a possible mechanism (which probably works in tandem with other mechanisms) to lower the

potentially high subduction velocities which are expected in plates with old and cold slabs. *We removed the phrase "(at least in part)"*

11. *[Line 156]: 1024 Pa×s – 1025 Pa×s should be changed to $10^{24}$ Pa×s – $10^{25}$ Pa×s.*
    *[Line 383]: Fig. B1 needs to be changed to Fig. C1.*
    *We revised the manuscript according to these comments*

[revised manuscript text omitted]

Table AS1:-1: Physical and rheological parameters of all models.

[Figure]

Appendix B, illustration S2. Illustration of the evolution of all models discussed in the main text

 Figure S2-1: Viscosity field evolution at various times, similar to Fig. 3, for the long-plate case (top row of each model) and short-plate case (bottom row of each model) for all models. White contour marks the 1100 °C isotherm used as the outline of the lithosphere. The vertical and horizontal scales are identical and only part of the full model domain is shown. t-t660

indicates the time since the initial interaction of the slab with the ULMB.

[Figure]

65  Figure B1: Evolution of all types of models (top – Long models, bottom – Short models)

**Strong lithosphere (factor 2)**

[Figure]

**Strong lithosphere (factor 10)**

[Figure]

Grey lines mark 220 km, 660 km (ULMB) and 1000 km depths.

[Figure]

 **Figure S2-1 (continued)**

[Figure]

**Figure** **S2-1 (continued**)

**Combined weak asthenosphere and strong lithosphere (factor 2)**

[Figure]

**Combined weak asthenosphere and strong lithosphere (factor 10)**

[Figure]

**Self consistent rheology (factor 2)**

[Figure]

**Reference**

Figure

[Figure]

**Reference**

[Figure]

80

Figure B2: Evolution graphs (as Fig. 4) for all models

**Weak asthenosphere**

[Figure]

**Strong lithosphere (factor 2)**

85 **S2-2: Temporal evolution, similar to Fig. 4, of the long-plate case (full lines) and the short-plate case (dashed lines) for all models. t-t660 indicates the time since the initial interaction of the slab with the ULMB. Panels show (from top left panel, along rows from top to bottom): (1) Velocity of the subducting plate (positive towards the upper plate, red) and the trench (positive away from the upper plate, yellow), measured at 2000 km distance from the initial subducting plate ridge (left hand boundary). (2) Displacement of the subducting plate (red) and the trench (yellow) relative to the initial condition. (3) Percent of plate convergence (calculated as the sum of trench retreat and plate displacement) achieved by trench retreat. (4) Upper-mantle slab pull and basal drag below the subducting plate, calculated as described in the main text. (5) Basal drag force from (4). (6) Ratio of basal drag to upper mantle**
90 **slab pull force.**

[Figure]

**Weak asthenosphere**

[Figure]

**Strong lithosphere (factor 2)**

[Figure]

Figure S2-2 (continued)

**Strong lithosphere (factor 10)**

[Figure]

**Combined weak asthenosphere and strong lithosphere (factor 2)**

[Figure]

**Combined weak asthenosphere and strong lithosphere (factor 10)**

[Figure]

**Self consistent rheology**

[Figure]

95

**Figure S2-2 (continued)**

[Figure]

Figure S2-3 (a) Temporal evolution of parameters used for the calculation of basal drag (see main text for further details), for

100    the long-plate reference model (left panels) and short-plate reference model (right panels). Top panel shows the velocities of

the lithosphere (red line) and asthenosphere (blue line) (measured along the dashed and dotted lines in the bottom panel,

respectively). Middle panel shows the magnitude of the viscosity of the asthenosphere (green line, measured along the dotted

line in the bottom panel). Bottom panel shows a vertical profile (0-660 km depth) of the magnitude of the viscosity field.

White lines mark the base of the lithosphere (1100°C isotherm) and asthenosphere (constant depth of 220 km). Parameters

105    for the lithosphere were measured at 20 km depth (black dashed line) and parameters for the asthenosphere at 160 km depth

(black dotted line). Time before the initial interaction of the slab with the ULMB, t660, is not shown in Fig. 4, 6 and 2.2 and

shaded in this figure. (b) Location of the vertical profile location along which the quantities in (a) are measured (brown line),

marked on an outline of the lithosphere at the initial condition of the long-plate reference model (magenta line) and the short-plate reference model (green line). The vertical and horizontal spatial scales are identical and only part of the full model

[Figure]

**Strong lithosphere (factor 10)**

[Figure]

**Combined weak asthenosphere and strong lithosphere (factor 2)**

[Figure]

110   domain is shown.

**Combined weak asthenosphere and strong lithosphere (factor 10)**

[Figure]

**Self consistent rheology**

[Figure]

Figure B2 (continue)

Appendix C,Grey lines mark 220 km, 660 km (ULMB) and 1000 km depth.

115

**S3. Cenozoic subduction zone parameters**

We used GPlates (Müller et al., 2018) with the plate reconstruction of Müller et al. (2016) to make an updated compilation of the average velocities, average age at the trench and size of the subducting plates for major subduction zones through the Cenozoic (0-60 Ma). This was done for all trenches previously studied by Sdrolias and Müller (2006) who considered the Andean subduction zone, Central-North Farallon subduction systems, subduction below Alaska and the Aleutians, subduction below Japan-Kuriles-Kamchatka, subduction below Izu-Bonin-Marianas, below Tonga-Kermadec and below the Sunda-Banda arc. To these we added the subduction of the Philippine Sea plate and the final stages of the subduction of the Izanagi and Kula plates. In processing this database, we considered all the Pacific subduction systems (i.e. Alaska-Aleutians, Japan-Kuriles-Kamchatka, Izu-Bonin-Marianas and Tonga-Kermadec) as a single Pacific system.

The relevant trenches were identified by extracting the global subducted segments of all plate polygons in GPlates. All trenches were sampled at 50 km intervals, and the coordinates were output and then plotted to select those belonging to the major subducting plate systems listed above. For each selected system, age and velocities along the trench were plotted every 10 Myr to check that all data made sense, and to remove edge points where these were clearly anomalous from the rest of the trench (e.g. because of an anomalous age or convergence direction). We evaluated the point velocity of the subducting plate, of the overriding plate as well as convergence velocity and direction to select the segments to analyse. Maps showing the trench segments included in our analysis at each stage can be found in Fig. S3-1, where the sampling points along the trenches are also coloured according to the subducting plate absolute velocity (in the moving hotspot reference frame of O'Neill et al., 2005) and age. These maps illustrate the evolving set of subduction systems through the Cenozoic as well as the variability of age and velocity along each trench. Aside from some edge points, we excluded from our analysis the part of the South American trench at 40 and 50 Myr which was south of the Antarctic ridge and the Cocos subduction system at 20 Ma, because according to the plate motion model there is limited convergence along most of the trench at this time. Finally, we considered for our analysis the mean value of the velocities and trench age for each subducting system, and the standard variation of the mean value as the uncertainty.

There is  some variation of both velocity and age along each trench, and each trench has a unique tectonic evolution. Nonetheless, the lack of a trend between velocity and size and the overall correlation between size and age are general features of the Cenozoic set of subduction zones, not dependent on including or excluding one or the other system, or on  different definitions of each system, or on consideration of the total subducting plate velocity or only the normal component of velocity.

[Figure]

**Figure S3-1: Evolution of  trenches considered in the _global_ analysis of  Cenozoic subduction systems, in intervals of 10 Myr. 3 maps are**

145

150

shown for each time interval, displaying the trenches used for the analysis (left), absolute subducting plate velocity in hotspot reference frame (O'Neill et al., 2005; middle) and age at the trench (right). Grey lines represent other boundaries of the subducting plates. The background light grey shows present day global coastlines for reference. Data is based on Müller et al. (2016) and processed using GPlates and Cartopy (Met Office, 2015; Müller et al., 2018).

[Figure]

[Figure]

[Figure]

**Figure C2:S3-2:** Velocity of subducting plates  as a function of plate size. Plate size as typical length
this calculated as the square root of the surface area of the plate
. (a) Cenozoic
dataset is based on plate reconstruction from Müller et al. (2016) and analysed using GPlates (Müller et al., 2018). Large markers
indicate present day values from this reconstruction. (b) Present day velocities from  Schellart et al.
(2007) and Sdrolias and Müller (2006), as a function of plate size
 are calculated in lower mantle (plume) reference frame (O'Neill et al., 2005).

---

## Referee Report (RR1)

The sentence below [Lines 11-13] may be removed for a better flow in the abstract:

*Furthermore, in single subduction system models, low basal drag, associated with a low ratio of asthenospheric to lithospheric viscosity, leads to subduction behaviour most consistent with the observation that trench migration velocities are generally low compared to convergence velocities.*

The information above may be shortened by adding the red section below on Line 17:

 … including low ratios of trench over plate motions that would be maintained at a high ratio of lithosphere to asthenosphere viscosity.

(I find going from lithosphere to asthenosphere is easier to grasp, similar to how you did on Line 453)

Lines [297-299] needs rewording.

*We also tested whether a self-consistent rheology that yields a stronger lithosphere and weaker asthenosphere than the reference models behaves similarly as the models where we artificially prescribed regions of modified viscosity.*

The Conclusion looks a bit wordy to me. You may consider shortening it by giving the key ideas in one or two paragraphs at most.

Ömer F. Bodur

---

## Author Response (AR2)

We thank the referee, **Ömer F. Bodur**, and the editors, **Susanne Buiter** and **Patrice Rey**, for their insights and comments. We considered each comment and revised the manuscript where necessary. Please see below our responses to each comment (in blue, with changes made in *italics*) and the annotated manuscript for the revisions.

Susanne Buiter:
- I understand that the parameters in the flow laws follow Garel et al 2014 and were tuned to fit geophysical observations. For those instances where the flow law parameters were either set or motivated by values of laboratory experiments, could you please mention the original references to these laboratory experiments?
  *We added references to the model set-up section (lines 139-145): "The parameterisation of diffusion and dislocation creep mechanisms are consistent with the experimental range derived from olivine deformation (e.g. Hirth and Kohlstedt, 2003) and predict realistic bulk upper mantle viscosities consistent with large-scale observations (Garel et al., 2020). The parameterisation of the Peierls creep is simplified (Garel et al., 2014), but is consistent with first order approximation of the dependency on temperature- and strain-rate (e.g. Kameyama et al., 1999). The yield stress mechanism is derived from Byerlee's law (Eq. S1.8) using rock-deformation experimental data (e.g. Burov, 2011). Such parameterisation of a low yield stress is common in geodynamical modelling (e.g. Mallard et al., 2016) to model surface plate deformation.". We also edited the caption of table S1-1 to include references to relevant experimental data.* Note these references also follow Garel et al., 2014.

Patrice Rey:
- I have noticed that in your paper you choose the base of the asthenosphere are 220km. Is this a commonly accepted definition for the asthenosphere? Reference?
  The depth of the asthenosphere was determined as the depth of transition from the low viscosity characterising the asthenosphere to the background viscosity of the upper mantle (see figure S2-3 for detailed evolution of the viscosity of the mantle). We refer to this in the 'model set up' section (lines 157-159): "We set the base of the asthenosphere at 220-km depth, which is below the bulk of the modelled minimum in viscosity, and consistent with the seismically imaged depth of the base of the asthenospheric low wave-speed zone (e.g. Dziewonski and Anderson, 1981; French et al., 2013)."
- In Table S1, the values of the pre-factor A are 1e-150 1e-145 to 1e-300, can you check these figures? I would have thought that most computers won't be able to deal with them.
  The values mentioned in Table S1-1 are for the prefactor of the Peierls creep mechanism. The viscosity is calculated in our models using eq. S1.6, which doesn't use the prefactor directly but rather the value of $A^{(-1/n)}$. Since n for Peierls creep is 20, the values stored in memory for the calculations are of the order of 7 to 15 and not -300 to -145. These values do not exceed the numerical limitations of most machines.
  *We revised the caption for table S1-1 to refer to these values (lines 49-50): "Lower mantle pre-factor value for dislocation and Peierls creep are set at a low value to ensure diffusion creep-controlled viscosity in the lower mantle (in accordance with eq. S1.5)."*

Ömer F. Bodur:
- The sentence below [Lines 11-13] may be removed for a better flow in the abstract:
  "Furthermore, in single subduction system models, low basal drag, associated with a low ratio of asthenospheric to lithospheric viscosity, leads to subduction behaviour most consistent with the observation that trench migration velocities are generally low compared to convergence velocities."

The information above may be shortened by adding the red section below on Line 17: "… including low ratios of trench over plate motions that would be maintained at a high ratio of lithosphere to asthenosphere viscosity."
(I find going from lithosphere to asthenosphere is easier to grasp, similar to how you did on Line 453)

*We revised the sentence in the abstract and removed the reference to viscosity ratio: "Furthermore, in models of single subduction system, low basal drag leads to subduction behaviour most consistent with the observation that trench migration velocities are generally low compared to convergence velocities."*

- Lines [297-299] needs rewording.
  "We also tested whether a self-consistent rheology that yields a stronger lithosphere and weaker asthenosphere than the reference models behaves similarly as the models where we artificially prescribed regions of modified viscosity."

  *We rephrased the sentence: "We tested how a model in which the flow law parameters were modified to produce a self-consistent rheology that yields a stronger lithosphere and weaker asthenosphere than the reference models compared to models with artificially prescribed regions of higher/lower viscosity"*

- The Conclusion looks a bit wordy to me. You may consider shortening it by giving the key ideas in one or two paragraphs at most.

  We considered this suggestion but felt that a longer conclusion that includes a short recap of the motivation is helpful for this paper.

[revised manuscript text omitted]

This supplement comprises three main sections and references section:

**S1.** Governing equations and rheology calculations for the numerical models

**S2.** Illustration of the evolution of all models discussed in the main text

**S3.** Cenozoic subduction zone parameters

**S1. Governing equations and rheology calculations for the numerical models**

We solve flow for incompressible Stokes fluid, under the Boussinesq approximation, assuming mass, momentum and energy conservation equations:

$$\partial_i u_i = 0 \tag{S1.1}$$

$$\partial_i \sigma_{ij} + \Delta\rho g_j = 0 \tag{S1.2}$$

$$\frac{\partial T}{\partial t} + u_i \partial_i T - \kappa \partial_i^2 T = 0 \tag{S1.3}$$

where $u$ is the velocity, $\sigma$ is the stress tensor, $g$ is gravity, $T$ is temperature, $\kappa$ is the thermal diffusivity and $\Delta\rho = -\alpha\rho_s\Delta T$ is the density difference due to temperature, with $\alpha$ the coefficient of thermal expansion, $\rho_s$ the reference (surface) mantle density and $\Delta T$ the difference in temperature from the surface.

Viscosity is therefore the ratio of deviatoric stress to strain rate:

$$\mu = \frac{\tau_{ij}}{2\dot{\varepsilon}_{ij}} = \frac{\sigma_{ij} + P\delta_{ij}}{2\dot{\varepsilon}_{ij}} \tag{S1.4}$$

where $\tau$ is the deviatoric stress, $\dot{\varepsilon}$ is the strain rate, $P$ is the dynamic pressure and $\delta_{ij}$ is the delta function. The viscosity is calculated in our models using:

$$\mu = \left(\frac{1}{\mu_{diff}} + \frac{1}{\mu_{disl}} + \frac{1}{\mu_y} + \frac{1}{\mu_{Pie}}\right)^{-1} \tag{S1.5}$$

where $\mu_{diff}$, $\mu_{disl}$, $\mu_y$ and $\mu_{Pie}$ are the viscosities calculated using diffusion creep, dislocation creep, yielding mechanism and simplified Pierels creep, respectively. The viscosities derived from diffusion, dislocation and Pierels creep are calculated
30  using the generalised equation:

$$\mu_{diff\backslash disl\backslash Pie} = A^{\frac{-1}{n}} \exp\left(\frac{E + P_l V}{nRT_{ad}}\right) \dot{\varepsilon}_{II}^{\frac{1-n}{n}} \tag{S1.6}$$

$$P_l = \rho g D \tag{S1.7}$$

where $A$ is a prefactor, $n$ is the stress exponent, $E$ and $V$ are the activation energy and volume, respectively, $P_l$ is the lithostatic pressure, $R$ the gas constant, $\dot{\varepsilon}_{II}$ is the second invariant of the strain rate tensor and $D$ is the depth. $T_{ad}$ is the
35  temperature adjusted with an adiabatic gradient of 0.5°K/km in the upper mantle and 0.3°K/km in the lower mantle (Fowler, 2005). The yielding mechanism is calculated as:

$$\mu_y = \frac{\tau_y}{2\dot{\varepsilon}_{II}} = \frac{\min(\tau_s + f_c P_l, \tau_{y\,max})}{2\dot{\varepsilon}_{II}} \tag{S1.8}$$

where $\tau_y$ is the yield stress, $\tau_s$ is the surface yield stress, $f_c$ is the friction coefficient and $\tau_{y\,max}$ is the maximum yield stress. The viscosity field is capped by both minimum and maximum values. The yielding viscosity is adjusted within the weak
40  decoupling layer by applying a different friction coefficient:

$$\mu_{y\,weak} = \frac{\min(\tau_s + f_{c\,weak} P_l, \tau_{y\,max})}{2\dot{\varepsilon}_{II}} \tag{S1.9}$$

The initial temperature field in the lithosphere is calculated using the half-space cooling equation from Turcotte and Schubert (2002):

$$T = T_s + \Delta T \cdot \mathrm{erf}\left(\frac{D}{2\sqrt{\kappa t_{age}}}\right) \tag{S1.10}$$

45  where $T_s$ is the surface temperature and $t_{age}$ is the age.

| Quantity | Symbol | Units | Value UM (Reference) [a] | Value UM (modified self-consistent) [b] | Value LM [a] |
|---|---|---|---|---|---|
| Gravity | $g$ | $m \cdot s^{-2}$ | 9.8 | | |
| Thermal expansivity coefficient | $\alpha$ | $K^{-1}$ | $3.0 \cdot 10^{-5}$ | | |
| Thermal diffusivity | $\kappa$ | $m^2 \cdot s^{-1}$ | $10^{-6}$ | | |
| Reference (surface) density | $\rho_s$ | $kg \cdot m^{-3}$ | 3300.0 | | |
| Cold, surface temperature | $T_s$ | $K$ | 273.0 | | |
| Hot, mantle temperature | $T_m$ | | 1573.0 | | |
| Gas constant | $R$ | $J \cdot K^{-1} \cdot mol^{-1}$ | 8.3145 | | |
| Maximum viscosity (Strong Lithosphere model) | $\mu_{max}$ | $Pa \cdot s$ | $10^{26}$ | | |
| Maximum viscosity (Reference and all other models) | | | $10^{25}$ | | |
| Minimum viscosity | $\mu_{min}$ | | $10^{18}$ | | |
| **Diffusion Creep** | | | | | |
| Activation energy | $E$ | $J \cdot mol^{-1}$ | $300.0 \cdot 10^3$ | $335.0 \cdot 10^3$ | $200.0 \cdot 10^3$ |
| Activation volume | $V$ | $m^3 \cdot mol^{-1}$ | $4.0 \cdot 10^{-6}$ | $5.0 \cdot 10^{-6}$ | $1.5 \cdot 10^{-6}$ |
| Pre-factor | $A$ | $Pa^{-n} \cdot s^{-1}$ | $3.0 \cdot 10^{-11}$ | $1.5 \cdot 10^{-9}$ | $6.0 \cdot 10^{-17}$ |
| Stress exponent | $n$ | | 1.0 | | |
| **Dislocation Creep (UM)** | | | | | |
| Activation energy | $E$ | $J \cdot mol^{-1}$ | $540.0 \cdot 10^3$ | $472.0 \cdot 10^3$ | $300.0 \cdot 10^3$ |
| Activation volume | $V$ | $m^3 \cdot mol^{-1}$ | $12.0 \cdot 10^{-6}$ | $11.0 \cdot 10^{-6}$ | $2.0 \cdot 10^{-6}$ |
| Pre-factor | $A$ | $Pa^{-n} \cdot s^{-1}$ | $5.0 \cdot 10^{-16}$ | $1.34 \cdot 10^{-17}$ | $10^{-42}$ |
| Stress exponent | $n$ | | 3.5 | 3.472 | 3.5 |
| **Peierls Creep (UM)** | | | | | |
| Activation energy | $E$ | $J \cdot mol^{-1}$ | $540.0 \cdot 10^3$ | $540.0 \cdot 10^3$ | $300.0 \cdot 10^3$ |
| Activation volume | $V$ | $m^3 \cdot mol^{-1}$ | $10.0 \cdot 10^{-6}$ | $10.0 \cdot 10^{-6}$ | $2.0 \cdot 10^{-6}$ |
| Pre-factor | $A$ | $Pa^{-n} \cdot s^{-1}$ | $10^{-150}$ | $10^{-145}$ | $10^{-300}$ |
| Stress exponent | $n$ | | 20.0 | | |
| **Yield Strength Law** | | | | | |
| Surface yield strength | $\tau_s$ | $MPa$ | 2.0 | | |
| Friction coefficient | $f_c$ | | 0.2 | | |
| Friction coefficient (decoupling layer) | $f_{c\,weak}$ | | 0.02 | 0.07 | 0.02 |
| Maximum yield strength | $\tau_{y\,max}$ | $MPa$ | 10,000 | | |

**Table S1-1: Physical and rheological parameters of all models, following the model set-up used by Garel et al., 2014. (a) Activation parameters and stress exponent are consistent with experimental data on olivine (e.g., Karato and Wu, 1993; Hirth and Kohlstedt, 1995; Ranalli, 1995; Hirth and Kohlstedt, 2003; Korenaga and Karato, 2008). Lower mantle pre-factor value for dislocation and Peierls creep are set at a low value to ensure diffusion creep-controlled viscosity in the lower mantle (in accordance with eq. S1.5).**

50

**(b) Upper mantle values for the modified self-consistent models were determined using optimisation method described in Maunder et al., 2016. See main text for further details**.

**S2. Illustration of the evolution of all models discussed in the main text**

[Figure]

55  **Figure S2-1: Viscosity field evolution at various times, similar to Fig. 3, for the long-plate case (top row of each model) and short-plate case (bottom row of each model) for all models. White contour marks the 1100 °C isotherm used as the outline of the**

**lithosphere. The vertical and horizontal scales are identical and only part of the full model domain is shown. t-t660 indicates the time since the initial interaction of the slab with the ULMB. Grey lines mark 220 km, 660 km (ULMB) and 1000 km depths.**

[Figure]

**Figure S2-1 (continued)**

[Figure]

**Figure S2-1** (continued)

**Self consistent rheology (factor 2)**

[Figure]

65    **Figure S2-1 (continued)**

**Reference**

**Figure S2-2: Temporal evolution, similar to Fig. 4, of the long-plate case (full lines) and the short-plate case (dashed lines) for all models. t-t660 indicates the time since the initial interaction of the slab with the ULMB. Panels show (from top left panel, along rows from top to bottom): (1) Velocity of the subducting plate (positive towards the upper plate, red) and the trench (positive away from the upper plate, yellow), measured at 2000 km distance from the initial subducting plate ridge (left hand boundary). (2) Displacement of the subducting plate (red) and the trench (yellow) relative to the initial condition. (3)**

[Figure]

**Percent of plate convergence (calculated as the sum of trench retreat and plate displacement) achieved by trench retreat. (4) Upper-mantle slab pull and basal drag below the subducting plate, calculated as described in the main text. (5) Basal drag force from (4). (6) Ratio of basal drag to upper mantle slab pull force.**

**Weak asthenosphere**

[Figure]

**Strong lithosphere (factor 2)**

[Figure]

**Figure S2-2 (continued)**

**Strong lithosphere (factor 10)**

[Figure]

**Combined weak asthenosphere and strong lithosphere (factor 2)**

[Figure]

**Figure S2-2 (continued)**

**Combined weak asthenosphere and strong lithosphere (factor 10)**

[Figure]

**Self consistent rheology**

[Figure]

 **Figure S2-2 (continued)**

[Figure]

**Figure S2-3 (a)** Temporal evolution of parameters used for the calculation of basal drag (see main text for further details), for the long-plate reference model (left panels) and short-plate reference model (right panels). Top panel shows the velocities of the lithosphere (red line) and asthenosphere (blue line) (measured along the dashed and dotted lines in the bottom panel, respectively). Middle panel shows the magnitude of the viscosity of the asthenosphere (green line, measured along the dotted line in the bottom panel). Bottom panel shows a vertical profile (0-660 km depth) of the magnitude of the viscosity field. White lines mark the base of the lithosphere (1100°C isotherm) and asthenosphere (constant depth of 220 km). Parameters for the lithosphere were measured at 20 km depth (black dashed line) and parameters for the asthenosphere at 160 km depth (black dotted line). Time before the initial interaction of the slab with the ULMB, $t_{660}$, is not shown in Fig. 4, 6 and 2.2 and shaded in this figure. **(b)** Location of the vertical profile location along which the quantities in (a) are measured (brown line), marked on an outline of the lithosphere at the initial condition of the long-plate reference model (magenta line) and the short-plate reference model (green line). The vertical and horizontal spatial scales are identical and only part of the full model domain is shown. Grey lines mark 220 km, 660 km (ULMB) and 1000 km depth.

**S3. Cenozoic subduction zone parameters**

115     We used GPlates (Müller et al., 2018) with the plate reconstruction of Müller et al. (2016) to make an updated compilation of the average velocities, average age at the trench and size of the subducting plates for major subduction zones through the Cenozoic (0-60 Ma). This was done for all trenches previously studied by Sdrolias and Müller (2006) who considered the Andean subduction zone, Central-North Farallon subduction systems, subduction below Alaska and the Aleutians, subduction below Japan-Kuriles-Kamchatka, subduction below Izu-Bonin-Marianas, below Tonga-Kermadec and below the

120     Sunda-Banda arc. To these we added the subduction of the Philippine Sea plate and the final stages of the subduction of the Izanagi and Kula plates. In processing this database, we considered all the Pacific subduction systems (i.e. Alaska-Aleutians, Japan-Kuriles-Kamchatka, Izu-Bonin-Marianas and Tonga-Kermadec) as a single Pacific system.

    The relevant trenches were identified by extracting the global subducted segments of all plate polygons in GPlates. All trenches were sampled at 50 km intervals, and the coordinates were output and then plotted to select those belonging to the

125     major subducting plate systems listed above. For each selected system, age and velocities along the trench were plotted every 10 Myr to check that all data made sense, and to remove edge points where these were clearly anomalous from the rest of the trench (e.g. because of an anomalous age or convergence direction). We evaluated the point velocity of the subducting plate, of the overriding plate as well as convergence velocity and direction to select the segments to analyse. Maps showing the trench segments included in our analysis at each stage can be found in Fig. S3-1, where the sampling points along the

130     trenches are also coloured according to the subducting plate absolute velocity (in the moving hotspot reference frame of O'Neill et al., 2005) and age. These maps illustrate the evolving set of subduction systems through the Cenozoic as well as the variability of age and velocity along each trench. Aside from some edge points, we excluded from our analysis the part of the South American trench at 40 and 50 Myr which was south of the Antarctic ridge and the Cocos subduction system at 20 Ma, because according to the plate motion model there is limited convergence along most of the trench at this time. Finally,

135     we considered for our analysis the mean value of the velocities and trench age for each subducting system, and the standard variation of the mean value as the uncertainty.

There is some variation of both velocity and age along each trench, and each trench has a unique tectonic evolution. Nonetheless, the lack of a trend between velocity and size and the overall correlation between size and age are general features of the Cenozoic set of subduction zones, not dependent on including or excluding one or the other system, or on different definitions of each system, or on consideration of the total subducting plate velocity or only the normal component of velocity.Figure S3-1: Evolution of velocities and ages along the trenches considered in the global analysis of Cenozoic subduction systems, in intervals of 10 Myr. 3 maps are shown for each time interval, displaying the trenches used for the analysis (left), absolute subducting plate velocity in hotspot reference frame (O'Neill et al., 2005; middle) and age at the trench (right). Grey lines represent other boundaries of the subducting plates. The background light grey shows present day global coastlines for reference. Data is based on Müller et al. (2016) and processed using GPlates and Cartopy (Met Office, 2015; Müller et al., 2018).

[Figure]

[Figure]

Figure S3-2: Velocity of subducting plates as a function of plate size. Plate size as typical length is calculated as the square root of the surface area of the plate. (a) Cenozoic dataset is based on plate reconstruction from Müller et al. (2016) and analysed using GPlates (Müller et al., 2018). Large markers indicate present day values from this reconstruction. (b) Present day velocities from Schellart et al. (2007) and Sdrolias and Müller (2006), as a function of plate size from Conrad and Hager (1999), same as shown in Fig. 1. All velocities are calculated in lower mantle (plume) reference frame (O'Neill et al., 2005).